# Projective Proximal Gradient Descent for a Class of Nonconvex Nonsmooth Optimization Problems: Fast Convergence Without Kurdyka-Łojasiewicz (KŁ) Property

**Yingzhen Yang**
School of Computing and Augmented Intelligence
Arizona State University, Tempe, AZ 85281, USA
yingzhen.yang@asu.edu

**Ping Li**
LinkedIn Ads
700 Bellevue Way NE, Bellevue, WA 98004, USA
pinli@linkedin.com

## Abstract

Nonconvex and nonsmooth optimization problems are important and challenging for statistics and machine learning. In this paper, we propose Projected Proximal Gradient Descent (PPGD) which solves a class of nonconvex and nonsmooth optimization problems, where the nonconvexity and nonsmoothness come from a nonsmooth regularization term which is nonconvex but piecewise convex. In contrast with existing convergence analysis of accelerated PGD methods for nonconvex and nonsmooth problems based on the Kurdyka-Łojasiewicz (KŁ) property, we provide a new theoretical analysis showing local fast convergence of PPGD. It is proved that PPGD achieves a fast convergence rate of $\mathcal{O}(1/k^2)$ when the iteration number $k \geq k_0$ for a finite $k_0$ on a class of nonconvex and nonsmooth problems under mild assumptions, which is locally Nesterov's optimal convergence rate of first-order methods on smooth and convex objective function with Lipschitz continuous gradient. Experimental results demonstrate the effectiveness of PPGD.

## 1 Introduction

Nonconvex and nonsmooth optimization problems are challenging ones which have received a lot of attention in statistics and machine learning (Bolte et al., 2014; Ochs et al., 2015). In this paper, we consider fast optimization algorithms for a class of nonconvex and nonsmooth problems presented as

$$\min_{\mathbf{x} \in \mathbb{R}^d} F(\mathbf{x}) = g(\mathbf{x}) + h(\mathbf{x}), \tag{1}$$

where $g$ is convex, $h(\mathbf{x}) = \sum_{j=1}^{d} h_j(\mathbf{x}_j)$ is a separable regularizer, each $h_j$ is piecewise convex. A piecewise convex function is defined in Definition 1.1. For simplicity of analysis we let $h_j = f$ for all $j \in [d]$, and $f$ is a piecewise convex function. Here $[d]$ is the set of natural numbers between 1 and $n$ inclusively. $f$ can be either nonconvex or convex, and all the results in this paper can be straightforwardly extended to the case when $\{h_j\}$ are different.

**Definition 1.1.** A univariate function $f \colon \mathbb{R} \to \mathbb{R}$ is piecewise convex if $f$ is lower semicontinuous and there exist intervals $\{\mathcal{R}_m\}_{m=1}^{M}$ such that $\mathbb{R} = \bigcup_{m=1}^{M} \mathcal{R}_m$, and $f$ restricted on $\mathcal{R}_m$ is convex for each $m \in [M]$. The left and right endpoints of $\mathcal{R}_m$ are denoted by $q_{m-1}$ and $q_m$ for all $m \in [M]$, where $\{q_m\}_{m=0}^{M}$ are the endpoints such that $q_0 = -\infty \leq q_1 < q_2 < \ldots < q_M = +\infty$. Furthermore, $f$ is either left continuous or right continuous at each endpoint $q_m$ for $m \in [M-1]$. $\{\mathcal{R}_m\}_{m=1}^{M}$ are also referred to as convex pieces throughout this paper.

It is important to note that for all $m \in [M-1]$, when $f$ is continuous at the endpoint $q_m$ or $f$ is only left continuous at $q_m$, $q_m \in \mathcal{R}_m$ and $q_m \notin \mathcal{R}_{m+1}$. If $f$ is only right continuous at $q_m$, $q_m \notin \mathcal{R}_m$ and $q_m \in \mathcal{R}_{m+1}$. This ensures that any point in $\mathbb{R}$ lies in only one convex piece.

When $M = 1$, $f$ becomes a convex function on $\mathbb{R}$, and problem (1) is a convex problem. We consider $M > 1$ throughout this paper, and our proposed algorithm trivially extends to the case when $M = 1$. We allow a special case that an interval $\mathcal{R}_m = \{q_m\}$ for $m \in [M-1]$ is a single-point set, in this case $q_{m-1} = q_m$. When there are no single-point intervals in $\{\mathcal{R}_m\}_{m=1}^{M}$, the minimum interval length is defined by $R_0 = \min_{m \in [M]} |\mathcal{R}_m|$. Otherwise, the minimum interval length is $R_0 = \min_{m \in [M]:\, |\mathcal{R}_m| \neq 0} |\mathcal{R}_m|$, where $|\cdot|$ denotes the length of an interval.

It is worthwhile to emphasize that the nonconvexity and nonsmoothness of many popular optimization problems come from piecewise convex regularizers. Below are three examples of piecewise convex functions with the corresponding regularizers which have been widely used in constrained optimization and sparse learning problems.

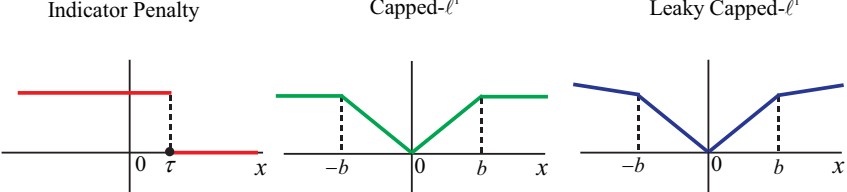

Figure 1: Illustration of three piecewise convex functions.

**Example 1.1.** (1) The indicator penalty function $f(x) = \lambda \mathbb{1}_{\{x < \tau\}}$ is piecewise convex with $\mathcal{R}_1 = (-\infty, \tau), \mathcal{R}_2 = [\tau, \infty)$. (2) The capped-$\ell^1$ penalty function $f(x) = f(x; \lambda, b) = \lambda \min\{|x|, b\}$ is piecewise convex with $\mathcal{R}_1 = (-\infty, -b], \mathcal{R}_2 = [-b, b], \mathcal{R}_3 = [b, \infty)$. (3) The leaky capped-$\ell^1$ penalty function (Wangni & Lin, 2017) $f = \lambda \min\{|x|, b\} + \beta \mathbb{1}_{\{|x| \geq b\}} |x - b|$ with $\mathcal{R}_1 = (-\infty, -b], \mathcal{R}_2 = [-b, b], \mathcal{R}_3 = [b, \infty)$. The three functions are illustrated in Figure 1. While not illustrated, $f(x) = \mathbb{1}_{\{x \neq 0\}}$ for the $\ell^0$-norm with $h(\mathbf{x}) = \|\mathbf{x}\|_0$ is also piecewise convex.

## 1.1 MAIN ASSUMPTION

The main assumption of this paper is that $g$ and $h$ satisfy the following conditions.

**Assumption 1** (Main Assumption). (a) $g$ is convex with $L$-smooth gradient, that is, $\|\nabla g(\mathbf{x}) - \nabla g(\mathbf{y})\|_2 \leq L_g \|\mathbf{x} - \mathbf{y}\|_2$. $F$ is coercive, that is, $F(\mathbf{x}) \to \infty$ when $\|\mathbf{x}\|_2 \to \infty$, and $\inf_{x \in \mathbb{R}^d} F(x) > -\infty$.

(b) $f\colon \mathbb{R} \to \mathbb{R}$ is a piecewise convex function and lower semicontinuous. Furthermore, there exists a small positive constant $s_0 < R_0$ such that $f$ is differentiable on $(q_m - s_0, q_m)$ and $(q_m, q_m + s_0)$ for all $m \in [M-1]$.

(c) The proximal mapping $\mathsf{prox}_{s f_m}$ has a closed-form solution for all $m \in [M]$, where $\mathsf{prox}_{s f_m}(x) := \mathrm{argmin}_{v \in \mathbb{R}} \frac{1}{2s} (v - x)^2 + f_m(v)$.

(d) $f$ has "negative curvature" at each endpoint $q_m$ where $f$ is continuous for all $m \in [M-1]$, that is, $\lim_{x \to q_m^-} f'(x) > \lim_{x \to q_m^+} f'(x)$. We define

$$C := \min_{m \in [M-1]:\, f \text{ continuous at } q_m} \left\{ \lim_{x \to q_m^-} f'(x) - \lim_{x \to q_m^+} f'(x) \right\} > 0. \tag{2}$$

In addition, $f$ has bounded Fréchet subdifferential, that is, $\sup_{x \in \bigcup_{m=1}^{M} \mathcal{R}_m^o} \sup_{v \in \tilde{\partial} f(x)} \|v\|_2 \leq F_0$ for some absolute constant $F_0 > 0$, where $\mathcal{R}^o$ denotes the interior of an interval.

Fréchet subdifferential is formally defined in Section 4. It is noted that on each $\mathcal{R}_m^o$, the convex differential of $f$ coincides with its Fréchet subdifferential. We define the minimum jump value of $f$ at noncontinuous endpoints by

$$J := \min \left\{ \min_{\substack{m \in [M-1]:\\ f \text{ is only right continuous at } q_m}} \left\{ \left| \lim_{y \to q_m^-} f(y) - f(q_m) \right| \right\}, \min_{\substack{m \in [M-1]:\\ f \text{ is only left continuous at } q_m}} \left\{ \left| \lim_{y \to q_m^+} f(y) - f(q_m) \right| \right\} \right\}. \tag{3}$$

Assumption 1 is mild and explained as follows. Smooth gradient in Assumption 1(a) is a commonly required in the standard analysis of proximal methods for non-convex problems, such as Bolte et al. (2014); Ghadimi & Lan (2016). The objective $F$ is widely assumed to be coercive in the nonconvex and nonsmooth optimization literature, such as Li & Lin (2015); Li et al. (2017). In addition, Assumption 1(b)-(d) are mild, and they hold for all the three piecewise convex functions in Example 1.1 as well as the $\ell^0$-norm.

It is noted that (1) covers a broad range of optimization problems in machine learning and statistics. The nonnegative programming problem $\min_{\mathbf{x} \in \mathbb{R}^d : \mathbf{x}_i \geq 0, i \in [d]} g(\mathbf{x})$ can be reduced to the regularized problem $g(\mathbf{x}) + h(\mathbf{x})$ with $f$ being the indicator penalty $f(x) = \lambda \mathbb{I}_{\{x < \tau\}}$ for a properly large $\lambda$. When $g$ is the squared loss, that is, $g(\mathbf{x}) = \|\mathbf{y} - \mathbf{D}\mathbf{x}\|_2^2$ where $\mathbf{y} \in \mathbb{R}^n$ and $\mathbf{D} \in \mathbb{R}^{n \times d}$ is the design matrix, (1) is the well-known regularized sparse linear regression problems with convex or nonconvex regularizers such as the capped-$\ell^1$ or $\ell^0$-norm penalty.

## 1.2 MAIN RESULTS AND CONTRIBUTIONS

We propose a novel method termed Projective Proximal Gradient Descent (PPGD), which extends the existing Accelerated Proximal Gradient descent method (APG) (Beck & Teboulle, 2009b; Nesterov, 2013) to solve problem (1). PPGD enjoys fast convergence rate by a novel projection operator and a new negative curvature exploitation procedure. Our main results are summarized below.

1. Using a novel and carefully designed projection operator and the Negative-Curvature-Exploitation algorithm (Algorithm 2), PPGD achieves a fast convergence rate for the nonconvex and nonsmooth optimization problem (1) which locally matches Nesterov's optimal convergence rate of first-order methods on smooth and convex objective function with Lipschitz continuous gradient. In particular, it is proved that under two mild assumptions, Assumption 1 and Assumption 2 to be introduced in Section 4, there exists a finite $k_0 \geq 1$ such that for all $k > k_0$,

$$F(\mathbf{x}^{(k)}) - F(\mathbf{x}^*) \leq \mathcal{O}(\frac{1}{k^2}), \tag{4}$$

where $\mathbf{x}^*$ is any limit point of $\left\{\mathbf{x}^{(k)}\right\}_{k \geq 1}$ lying on the same convex pieces as $\left\{\mathbf{x}^{(k)}\right\}_{k > k_0}$. Details are deferred to Theorem 4.4 in Section 4. It should be emphasized that this is the same convergence rate as that of regular APG on convex problems (Beck & Teboulle, 2009b;a).

2. Our analysis provides insights into accelerated PGD methods for a class of challenging nonconvex and nonsmooth problems. In contrast to most existing accelerated PGD methods (Li & Lin, 2015; Li et al., 2017) which employ the Kurdyka-Łojasiewicz (KŁ) property to analyze the convergence rates, PPGD provides a new perspective of convergence analysis without the KŁ property while locally matching Nesterov's optimal convergence rate. Our analysis reveals that the objective function makes progress, that is, its value is decreased by a positive amount, when the iterate sequence generated by PPGD transits from one convex piece to another. Such observation opens the door for future research in analyzing convergence rate of accelerated proximal gradient methods without the KŁ property.

It should be emphasized that the conditions in Assumption 1 can be relaxed while PPGD still enjoys the same order of convergence rate. First, the assumption that $f$ is piecewise convex can be relaxed to a weaker one to be detailed in Remark 4.6. Second, the proximal mapping in Assumption 1 does not need to have a closed-form solution.

Assuming that both $g$ and $h$ satisfy the KŁ Property defined in Definition A.1 in Section A.1 of the supplementary, the accelerated PGD algorithms in Li & Lin (2015); Li et al. (2017) have linear convergence rate with $\theta \in [1/2, 1)$, and sublinear rate $\mathcal{O}(k^{-\frac{1}{1-2\theta}})$ with $\theta \in (0, 1/2)$, where $\theta$ is the Kurdyka-Lojasiewicz (KŁ) exponent and both rates are for objective values.

To the best of our knowledge, most existing convergence analysis of accelerated PGD methods for nonconvex and nonsmooth problems, where the nonconvexity and nonsmoothness are due to the regularizer $h$, are based on the KŁ property. While (Ghadimi & Lan, 2016) provides analysis of an accelerated PGD method for nonconvex and nonsmooth problems, the nonconvexity comes from the smooth part of the objective function ($g$ in our objective function), so it cannot handle the problems discussed in this paper. Furthermore, the fast PGD method by Yang & Yu (2019) is restricted to $\ell^0$-

regularized problems. As a result, it remains an interesting and important open problem regarding the convergence rate of accelerated PGD methods for problem (1).

Another line of related works focuses on accelerated gradient methods for smooth objective functions. For example, Jin et al. (2018) proposes perturbed accelerated gradient method which achieves faster convergence by decreasing a quantity named Hamiltonian, which is the sum of the current objective value and the scaled squared distance between consecutive iterates, by exploiting the negative curvature. The very recent work Li & Lin (2022) further removes the polylogarithmic factor in the time complexity of Jin et al. (2018) by a restarting strategy.

Because our objective is nonsmooth, the Hamiltonian cannot be decreased by exploiting the negative curvature of the objective in the same way as Jin et al. (2018). However, we still manage to design a "Negative-Curvature-Exploitation" algorithm which decreases the objective value by an absolute positive amount when the iterates of PPGD cross endpoints. Our results are among the very few results in the optimization literature about fast optimization methods for nonconvex and nonsmooth problems which are not limited to $\ell^0$-regularized problems.

## 1.3 NOTATIONS

Throughout this paper, we use bold letters for matrices and vectors, regular lower letters for scalars. The bold letter with subscript indicates the corresponding element of a matrix or vector. $\|\cdot\|_p$ denotes the $\ell_p$-norm of a vector, or the $p$-norm of a matrix. $|\mathbf{A}|$ denotes the cardinality of a set $\mathbf{A}$. When $\mathcal{R} \subseteq \mathbb{R}$ is an interval, $\overline{\mathcal{R}}$ is the closure of $\mathcal{R}$, $\mathcal{R}^+$ denotes the set of all the points in $\mathbb{R}$ to the right of $\mathcal{R}$ while not in $\mathcal{R}$, and $\mathcal{R}^-$ is defined similarly denoting the set of all the points to the left of $\mathcal{R}$. The domain of any function $u$ is denoted by $\mathbf{dom}u$. $\lim_{x \to a^+}$ and $\lim_{x \to a^-}$ denote left limit and right limit at a point $a \in \mathbb{R}$. $\mathbb{N}$ is the set of all natural numbers.

## 2 ROADMAP TO FAST CONVERGENCE BY PPGD

Two essential components of PPGD contribute to its fast convergence rate. The first component, a combination of a carefully designed Negative-Curvature-Exploitation algorithm and a new projection operator, decreases the objective function by a positive amount when the iterates generated by PPGD transit from one convex piece to another. As a result, there can be only a finite number of such transitions. After finite iterations, all iterates must stay on the same convex pieces. Restricted on these convex pieces, problem (1) is convex. The second component, which comprises $M$ surrogate functions, naturally enables that after iterates reach their final convex pieces, they are operated in the same way as a regular APG does so that the convergence rate of PPGD locally matches Nesterov's optimal rate achieved by APG. Restricted on each convex piece, the piecewise convex function $f$ is convex. Every surrogate function is designed to be an extension of this restricted function to the entire $\mathbb{R}$, and PPGD performs proximal mapping only on the surrogate functions.

## 3 ALGORITHMS

Before presenting the proposed algorithm, we first define proximal mapping, and then describe how to build surrogate functions for each convex piece.

**Definition 3.1** (Proximal Mapping). The proximal mapping associated with function $u$ is defined as $\mathsf{prox}_u(\mathbf{x}) := \mathrm{argmin}_{\mathbf{v} \in \mathbb{R}^d} u(\mathbf{v}) + \frac{1}{2}\|\mathbf{v} - \mathbf{x}\|_2^2$ for $\mathbf{x} \in \mathbb{R}^d$, and $\mathsf{prox}_u(\cdot)$ is called the proximal mapping associated with $u$.

### 3.1 CONSTRUCTION OF THE SURROGATE FUNCTIONS $\{f_m\}_{m=1}^M$

Given that $f$ is convex on each convex piece $\mathcal{R}_m$, we describe how to construct a surrogate function $f_m \colon \mathbb{R} \to \mathbb{R}$ such that $f_m(x) = f(x)$ for all $x \in \mathcal{R}_m$. The key idea is to extend the domain of $f_m$ from $\mathcal{R}_m$ to $\mathbb{R}$ with the simplest structure, that is, $f_m$ is linear outside of $\mathcal{R}_m$. More concretely, if the right endpoint $q = q_m$ is not $+\infty$ and $f$ is continuous at $q$, then $f_m$ extends $f|_{\mathcal{R}_m}$ such that $f_m$ on $(q, +\infty)$ is linear. Similar extension applies to the case when $q = q_{m-1}$ is the left endpoint of

$\mathcal{R}_m$. Formally, we let

$$f_m(x) = \begin{cases} f(q_m) + v^-(x - q_m) & f \text{ is continuous at } q_m, \\ \lim_{y \to q_m^-} f(y) + v^-(x - q_m) & f(q_m) < \lim_{y \to q_m^-} f(y) \\ \lim_{y \to q_m^+} f(y) & f(q_m) < \lim_{y \to q_m^+} f(y), \end{cases} \tag{5}$$

for all $x \in \mathcal{R}_m^+$ if $q_m \neq +\infty$, where $v^- = \lim_{x \to q_m^-} f'(x)$. The surrogate function $f_m$ is defined similarly on $\mathcal{R}_m^-$ by

$$f_m(x) = \begin{cases} f(q_{m-1}) + v^+(x - q_{m-1}) & f \text{ is continuous at } q_{m-1}, \\ \lim_{y \to q_{m-1}^+} f(y) + v^+(x - q_{m-1}) & f(q_{m-1}) < \lim_{y \to q_{m-1}^+} f(y) \\ \lim_{y \to q_{m-1}^-} f(y) & f(q_{m-1}) < \lim_{y \to q_{m-1}^-} f(y), \end{cases} \tag{6}$$

for all $x \in \mathcal{R}_m^-$ if $q_{m-1} \neq -\infty$, where $v^+ = \lim_{x \to q_{m-1}^+} f'(x)$. Figure 2 illustrates the surrogate function $f_m(x)$ with $x \in \mathcal{R}_m^+$ for the three different cases in (5).

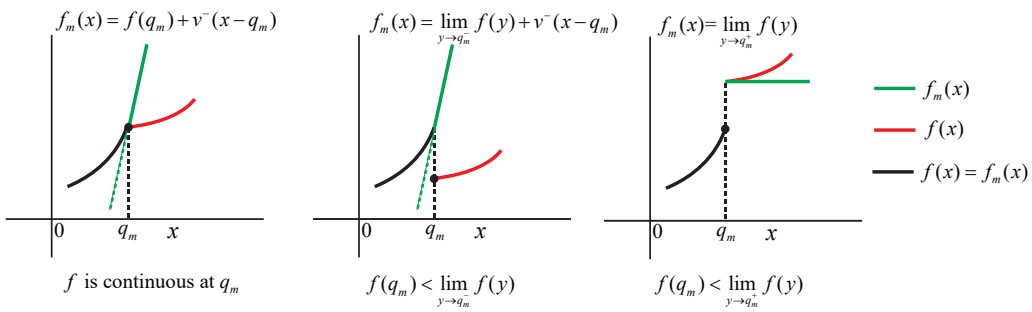

Figure 2: Illustration of three piecewise convex functions.

Example 3.1 explains how to build surrogate functions for two popular piecewise convex functions in the optimization literature. It can be observed from the definition of surrogate functions and Figure 2 that surrogate functions always have a simpler geometric structure than the original $f$, so it is usually the case that proximal mapping associated with the surrogate functions have closed-form expressions, given that proximal mapping associated with $f$ has a closed-form solution. The proximal mappings associated with the surrogate functions are also given in Example 3.1.

**Example 3.1** (Examples of Surrogate Functions). (1) The capped-$\ell^1$ penalty function $f(x) = \lambda \min\{|x|, b\}$ has three surrogate functions as $f_1 = f_3 \equiv \lambda b$, and $f_2 = \lambda |x|$. The proximal mappings for all $\{f_i\}_{i=1}^3$ has closed-form expressions, with $\text{prox}_{sf_1}(x) = \text{prox}_{sf_3}(x) = x$, and $\text{prox}_{sf_2}(x) = (1 - \lambda s/|x|)_+ x$ being the soft thresholding operator, where $(a)_+ = a$ when $a \geq 0$ and $(a)_+ = 0$ when $a < 0$. (2) The indicator penalty function $f(x) = \lambda \mathbb{1}_{\{x < \tau\}}$ has surrogate functions $f_1 \equiv \lambda$, $f_2 = f$. The proximal mappings are $\text{prox}_{sf_1}(x) = x$, $\text{prox}_{sf_2}(x) = \tau$ when $\tau - \sqrt{2\lambda s} < x < \tau$, and $\text{prox}_{sf_2}(x) = x$ otherwise except for the point $x = \tau - \sqrt{2\lambda s}$. While $\text{prox}_{sf_2}(x)$ has two values at $x = \tau - \sqrt{2\lambda s}$, it will not be evaluated at $x = \tau - \sqrt{2\lambda s}$ in our PPGD algorithm to be introduced.

## 3.2 Projective Proximal Gradient Descent

We now define a basic operator $P$ which returns the index of the convex piece that the input lies in, that is, $P(x) = m$ for $x \in \mathcal{R}_m$. It is noted that $P(x)$ is a single-valued function, so $P(x) \neq P(y)$ indicates $x, y$ are on different convex pieces, that is, there is no convex piece which contain both $x$ and $y$. Given a vector $\mathbf{x} \in \mathbb{R}^d$, we define $P(\mathbf{x}) \in \mathbb{R}^d$ such that $[P(\mathbf{x})]_i = P(\mathbf{x}_i)$ for all $i \in [d]$. We then introduce a novel projection operator $\mathbf{P}_{\mathbf{x}, R_0}(\cdot)$ for any $\mathbf{x} \in \mathbb{R}^d$. That is, $\mathbf{P}_{\mathbf{x}, R_0} : \mathbb{R}^d \to \mathbb{R}^d$, and $\mathbf{P}_{\mathbf{x}, R_0}(\mathbf{u})$ for all $\mathbf{u} \in \mathbb{R}^d$ is defined as

$$[\mathbf{P}_{\mathbf{x}, R_0}(\mathbf{u})]_i := \text{argmin}_{x \in \overline{\mathcal{R}_{P(\mathbf{x}_i)}} \cap B(\mathbf{x}_i, R_0)} |x - \mathbf{u}_i| \tag{11}$$

---

**Algorithm 1** Projected Proximal Gradient Descent $\left(\mathbf{x}^{(0)}\right)$

---

1: $\mathbf{z}^{(1)} = \mathbf{x}^{(1)} = \mathbf{x}^{(0)}$, $t_0 = 0$, $t_1 = 1$, endpoint $\{q_m\}_{m=1}^{M-1}$, step size $s$, constant $w_0 \in (0,1]$.
2: **for** $k = 1, \ldots,$ **do**

$$\mathbf{u}^{(k)} = \mathbf{x}^{(k)} + \frac{t_{k-1}}{t_k}(\mathbf{z}^{(k)} - \mathbf{x}^{(k)}) + \frac{t_{k-1} - 1}{t_k}(\mathbf{x}^{(k)} - \mathbf{x}^{(k-1)}) \tag{7}$$

$$\mathbf{w}^{(k)} = \mathbf{P}_{\mathbf{x}^{(k)}, R_0}(\mathbf{u}^{(k)}) \tag{8}$$

**for** $i = 1, \ldots, d,$ **do**

$$\mathbf{z}_i^{(k+1)} = \mathsf{prox}_{sf_{P(\mathbf{x}_i^{(k)})}}\left(\left[\mathbf{w}^{(k)} - s\nabla g(\mathbf{w}^{(k)})\right]_i\right) \tag{9}$$

$$t_{k+1} = \frac{\sqrt{1 + 4t_k^2} + 1}{2}.$$

**if** $F_{P(\mathbf{x}^{(k)})}(\mathbf{z}^{(k+1)}) \leq F(\mathbf{x}^{(k)})$ **then**

$$\mathbf{x}^{(k+1)} = \text{Negative-Curvature-Exploitation}\left(\mathbf{x}^{(k)}, \mathbf{z}^{(k+1)}, w_0\right) \tag{10}$$

**else**

$$\mathbf{x}^{(k+1)} = \mathbf{x}^{(k)}$$

---

for all $i \in [d]$, where $B(\mathbf{x}_i, R_0) \coloneqq [\mathbf{x}_i - R_0, \mathbf{x}_i + R_0]$. Since $\overline{\mathcal{R}_{P(\mathbf{x}_i)}} \cap B(\mathbf{x}_i, R_0)$ is a closed convex set, (11) is the projection of $\mathbf{u}_i$ onto the closed convex set $\overline{\mathcal{R}_{P(\mathbf{x}_i)}} \cap B(\mathbf{x}_i, R_0)$ which is well-defined. $\mathbf{P}_{\mathbf{x}, R_0}(\mathbf{u})$ can be computed very efficiently by setting the endpoint of $\mathcal{R}_{P(\mathbf{x}_i)} \cap B(\mathbf{x}_i, R_0)$ closer to $\mathbf{u}_i$ to $[\mathbf{P}_{\mathbf{x}, R_0}(\mathbf{u})]_i$.

---

**Algorithm 2** Negative-Curvature-Exploitation $\left(\mathbf{x}^{(k)}, \mathbf{z}^{(k+1)}, w_0\right)$

---

1: **if** $P(\mathbf{z}^{(k+1)}) = P(\mathbf{x}^{(k)})$ **then**
2:    $\mathbf{x}^{(k+1)} = \mathbf{z}^{(k+1)}$
3:    **return** $\mathbf{x}^{(k+1)}$
4: Flag = false
5: $\mathbf{z}' = \mathbf{z}^{(k+1)}$
6: **for** $i = 1, \ldots, d$ **do**
7:    **if** $P(\mathbf{z}_i^{(k+1)}) \neq P(\mathbf{x}_i^{(k)}) = m_i$ **then**
8:       **if** $f$ is continuous at $q(\mathbf{w}_i^{(k)})$ and $d_{i,1} \geq w_0 d_{i,0}$ **then**
9:          Flag = true
10:      **if** $f$ is not continuous at $q(\mathbf{w}_i^{(k)})$ **then**
11:         Flag = true
12:         **if** $\mathcal{R}_{P(\mathbf{z}_i^{(k+1)})} = \left\{q(\mathbf{w}_i^{(k)})\right\}$ **then**
13:            $\mathbf{z}_i' = q(\mathbf{w}_i^{(k)})$
14: **if** Flag = false **then**
15:    $\mathbf{x}^{(k+1)} = \mathbf{x}^{(k)}$
16: **else**
17:    $\mathbf{x}^{(k+1)} = \mathbf{z}'$
18: **return** $\mathbf{x}^{(k+1)}$

---

The Projected Proximal Gradient Descent (PPGD) algorithm is described in Algorithm 1. The following functions and quantities are defined which are useful for PPGD. Let $q(\mathbf{w}_i^{(k)})$ be the endpoint closest to $\mathbf{w}_i^{(k)}$ which lies in $[\mathbf{w}_i^{(k)}, \mathbf{z}_i^{(k+1)}]$ or $[\mathbf{z}_i^{(k+1)}, \mathbf{w}_i^{(k)}]$ when $P(\mathbf{z}_i^{(k+1)}) \neq P(\mathbf{x}_i^{(k)})$ at iteration $k$ of Algorithm 1. We define $F_{P(\mathbf{x})}(\mathbf{v}) \coloneqq g(\mathbf{v}) + \sum_{i=1}^d f_{P(\mathbf{x}_i)}(\mathbf{v}_i)$, $d_{i,0} \coloneqq \left|\mathbf{z}_i^{(k+1)} - \mathbf{w}_i^{(k)}\right|$, $d_{i,1} \coloneqq \left|\mathbf{z}_i^{(k+1)} - q(\mathbf{w}_i^{(k)})\right|$.

The idea of PPGD is to use $\mathbf{z}^{(k+1)}$ as a probe for the next convex pieces that the current iterate $\mathbf{x}^{(k)}$ should transit to. Compared to the regular APG described in Algorithm 4 in Section A.2 of the supplementary, the projection of $\mathbf{u}^{(k)}$ onto a closed convex set is used to compute $\mathbf{z}^{(k+1)}$. This is to make sure that $\mathbf{u}^{(k)}$ is "dragged" back to the convex pieces of $\mathbf{x}^{(k)}$, and the only variable that can explore new convex pieces is $\mathbf{z}^{(k+1)}$. We have a novel Negative-Curvature-Exploitation (NCE) algorithm described in Algorithm 2 which decides if PPGD should update the next iterate $\mathbf{x}^{(k+1)}$. In particular, if $\mathbf{z}^{(k+1)}$ is on the same convex pieces as $\mathbf{x}^{(k)}$, then $\mathbf{x}^{(k+1)}$ is updated to $\mathbf{z}^{(k+1)}$ only if $\mathbf{z}^{(k+1)}$ has smaller objective value. Otherwise, the NCE algorithm carefully checks if any one of the two sufficient conditions can be met so that the objective value can be decreased. If this is the case, then Flag is set to True and $\mathbf{x}^{(k+1)}$ is updated by $\mathbf{z}^{(k+1)}$, which indicates that $\mathbf{x}^{(k+1)}$ transits to convex pieces different from that of $\mathbf{x}^{(k)}$. Otherwise, $\mathbf{x}^{(k+1)}$ is set to the previous value $\mathbf{x}^{(k)}$. The two sufficient conditions are checked in line 8 and 10 of Algorithm 2. Such properties of NCE along with the convergence of PPGD will be proved in the next section.

## 4 CONVERGENCE ANALYSIS

In this section, we present the analysis for the convergence rate of PPGD in Algorithm 1. Before that, we present the definition of Fréchet subdifferential which is used to define critical points of the objective function.

**Definition 4.1.** (Fréchet subdifferential and critical points (Rockafellar & Wets, 2009)) Given a non-convex function $u\colon \mathbb{R}^d \to \mathbb{R} \cup \{+\infty\}$ which is a proper and lower semicontinuous function.

- For a given $\mathbf{x} \in \mathbf{dom}\,u$, its Fréchet subdifferential of $u$ at $\mathbf{x}$, denoted by $\tilde{\partial} u(\mathbf{x})$, is the set of all vectors $\mathbf{u} \in \mathbb{R}^d$ which satisfy

$$\liminf_{\mathbf{y}\neq\mathbf{x},\mathbf{y}\to\mathbf{x},\mathbf{y}\in\mathbf{dom}\,u} \frac{u(\mathbf{y}) - u(\mathbf{x}) - \langle \mathbf{u}, \mathbf{y} - \mathbf{x}\rangle}{\|\mathbf{y} - \mathbf{x}\|} \geq 0.$$

- The limiting subdifferential of $u$ at $\mathbf{x} \in \mathbb{R}^d$, denoted by written $\partial u(\mathbf{x})$, is defined by

$$\partial u(\mathbf{x}) = \{\mathbf{v} \in \mathbb{R}^d \mid \exists \tilde{\mathbf{x}}^k \to \mathbf{x}, u(\mathbf{x}^k) \to u(\mathbf{x}), \tilde{\mathbf{v}}^k \in \tilde{\partial} u(\mathbf{x}^k) \to \mathbf{v}\}.$$

The point $\mathbf{x}$ is a critical point of $u$ if $\mathbf{0} \in \partial u(\mathbf{x})$.

The following assumption is useful for our analysis when the convex pieces have one or more endpoints at which $f$ is continuous.

**Assumption 2** (Nonvanishing Gradient at Continuous Endpoints)**.** Let $q_m$ with $m \in [M-1]$ be an endpoint at which $f$ is continuous. There exists a positive constant $\varepsilon_0$ such that the following two conditions hold. If $q_m$ is a left endpoint, then $\inf_{\mathbf{x}\in\mathbb{R}^d, \mathbf{x}_i=q_m} |[\nabla g(\mathbf{x})]_i + p^+| \geq \varepsilon_0$ for all $i \in [d]$. If $q_m$ is a right endpoint, then $\inf_{\mathbf{x}\in\mathbb{R}^d, \mathbf{x}_i=q_m} |[\nabla g(\mathbf{x})]_i + p^-| \geq \varepsilon_0$. Here $p^+ := \lim_{y\to q_m^+} f'(y)$, and $p^- := \lim_{y\to q_m^-} f'(y)$.

We note that Assumption 2 is rather mild. For example, when $f(x) = \lambda \min\{|x|, b\}$ is the capped-$\ell^1$ penalty, then Assumption 2 holds when $\lambda > G$ and $\varepsilon_0$ can be set to $\lambda - G$. Such requirement for large $\lambda$ is commonly used for analysis of consistency of sparse linear models, such as Loh (2017). In addition, when there are no endpoints at which $f$ is continuous, then Assumption 2 is not required for the provable convergence rate of PPGD. Details are deferred to Remark 4.5.

We then have the important Theorem 4.3, which directly follows from Lemma 4.1 and Lemma 4.2 below. Theorem 4.3 states that the Negative-Curvature-Exploitation algorithm guarantees that, if convex pieces change across consecutive iterates $\mathbf{x}^{(k)}$ and $\mathbf{x}^{(k+1)}$, then the objective value is decreased by a positive amount. Before presenting these results, we note that the level set $\mathcal{L} := \overline{\{\mathbf{x} \mid F(\mathbf{x}) \leq F(\mathbf{x}^{(0)})\}}$ is bounded because $F$ is coercive. Since $\nabla g$ is smooth, we define the supremum of $\|\nabla g\|_2$ over an enlarged version of $\mathcal{L}$ as $G$:

$$G := \sup_{\mathbf{x}\in\mathcal{L}_{R_0}} \|\nabla g(\mathbf{x})\|_2, \quad \text{where } \mathcal{L}_{R_0} := \left\{\mathbf{x} \in \mathbb{R}^d \,\middle|\, \sup_{\mathbf{x}'\in\mathcal{L}} \|\mathbf{x} - \mathbf{x}'\|_2 \leq R_0\right\}. \qquad (12)$$

$G$ is finite due to the fact that $\mathcal{L}_{R_0}$ is compact, and it will serve as the upper bound for $\left\|\nabla g(\mathbf{w}^{(k)})\right\|_2$.

**Lemma 4.1** (Decrease of the objective function when convex pieces change). Define $A := L_g(G + \sqrt{d}F_0)$, $\kappa_0 := \min\{\kappa_1, \kappa_2\}$ with $\kappa_1 := sCw_0(\varepsilon_0 - sL_g(1-w_0)(G+F_0))$, $\kappa_2 := J - 2sF_0(G+F_0)$, and $\kappa := \kappa_0 - s\left(sL_g(G + \sqrt{d}F_0) + G\right)(G + \sqrt{d}F_0)$. Let the step size satisfy

$$s < s_1 := \min\left\{\frac{s_0}{G+F_0}, \frac{\varepsilon_0}{L_g(1-w_0)(G+F_0)}, \frac{J}{F_0G+G^2/2}, \frac{J}{2F_0(G+F_0)}, \frac{-G + \sqrt{G^2 + \frac{4A\kappa_0}{G+\sqrt{d}F_0}}}{2A}\right\},$$
(13)

and suppose $\left\|\nabla g(\mathbf{w}^{(k)})\right\|_2 \leq G$. If $P(\mathbf{x}^{(k+1)}) \neq P(\mathbf{x}^{(k)})$ for some $k \geq 1$ in the sequence $\left\{\mathbf{x}^{(k)}\right\}_{k\geq 1}$ generated by the PPGD algorithm, then under Assumption 1 and Assumption 2,

$$F(\mathbf{x}^{(k+1)}) \leq F(\mathbf{x}^{(k)}) - \kappa.$$
(14)

**Lemma 4.2.** The sequences $\left\{\mathbf{x}^{(k)}\right\}_{k\geq 1}$ and $\left\{\mathbf{w}^{(k)}\right\}_{k\geq 1}$ generated by the PPGD algorithm described in Algorithm 1 satisfy

$$F(\mathbf{x}^{(k)}) \leq F(\mathbf{x}^{(k-1)}), \quad \left\|\nabla g(\mathbf{w}^{(k)})\right\|_2 \leq G, \quad \forall k \geq 1.$$
(15)

**Theorem 4.3.** Suppose the step size $s < s_1$ where $s_1$ is defined in (13). If $P(\mathbf{x}^{(k+1)}) \neq P(\mathbf{x}^{(k)})$ for some $k \geq 1$ in the sequence $\left\{\mathbf{x}^{(k)}\right\}_{k\geq 1}$ generated by the PPGD algorithm, then under Assumption 1 and Assumption 2,

$$F(\mathbf{x}^{(k+1)}) \leq F(\mathbf{x}^{(k)}) - \kappa,$$
(16)

where $\kappa$ is defined in Lemma 4.1.

Due to Theorem 4.3, there can only be a finite number of $k$'s such that the convex pieces change across consecutive iterates $\mathbf{x}^{(k)}$ and $\mathbf{x}^{(k+1)}$. As a result, after finite iterations all the iterates generated by PPGD lie on the same convex pieces, so the nice properties of convex optimization apply to these iterates. Formally, we have the following theorem stating the convergence rate of PPGD.

**Theorem 4.4** (Convergence rate of PPGD). Suppose the step size $s < \min\left\{s_1, \frac{\varepsilon_0}{L_g(G+\sqrt{d}F_0)}, \frac{1}{L_g}\right\}$ with $s_1$ defined by (13), Assumption 1 and Assumption 2 hold. Then there exists a finite $k_0 \geq 1$ such that the following statements hold. (1) $P(\mathbf{x}^{(k)}) = \mathbf{m}^*$ for some $\mathbf{m}^* \in \mathbb{N}^d$ for all $k > k_0$. (2) Let $\Omega := \left\{\mathbf{x} \mid \mathbf{x} \text{ is a limit point of } \left\{\mathbf{x}^{(k)}\right\}_{k\geq 1}, P(\mathbf{x}) = \mathbf{m}^*\right\}$ be the set of all limit points of the sequence $\left\{\mathbf{x}^{(k)}\right\}_{k\geq 1}$ generated by Algorithm 1 lying on the convex pieces indexed by $\mathbf{m}^*$. Then for any $\mathbf{x}^* \in \Omega$, $F(\mathbf{x}^*) = \inf_{\{\mathbf{x} \in \mathbb{R}^d \mid P(\mathbf{x}) = \mathbf{m}^*\}} F(\mathbf{x})$, and

$$F(\mathbf{x}^{(k)}) - F(\mathbf{x}^*) \leq \frac{4}{k^2} U^{(k_0)}$$
(17)

for all $k > k_0$, where $U^{(k_0)} := \left(\frac{1}{2s}\|(t_{k_0-1}-1)\mathbf{x}^{(k_0-1)} - t_{k_0-1}\mathbf{z}^{(k_0)} + \mathbf{x}^*\|_2^2 + t_{k_0-1}^2(F(\mathbf{x}^{(k_0)}) - F(\mathbf{x}^*))\right)$. Moreover, if $f_{\mathbf{m}^*}$ does not take the third case in (5) or (6) for all $i \in [d]$, then $\mathbf{x}^*$ is a critical point of $F$, that is, $0 \in \partial F(\mathbf{x}^*)$.

**Remark 4.5.** If the convex pieces $\{\mathcal{R}_m\}_{m=1}^M$ has no endpoints at which $f$ is continuous, then Theorem 4.4 holds without requiring Assumption 2. In particular, Theorem 4.4 holds without Assumption 2 for problem (1) with indicator penalty or $\ell^0$-norm regularizer.

**Remark 4.6.** All the statements of Theorem 4.4 still hold if $f$ is not piecewise convex while satisfying all the other conditions of Assumption 1 and Assumption 2, if $f$ restricted on the final convex piece $\mathcal{R}_{\mathbf{m}_i^*}$ is convex for all $i \in [d]$. The reason is that we only need the convexity of $f$ on the final convex pieces to prove Nesterov's optimal convergence rate in (17).

**Roadmap of the proof of Theorem 4.4.** Proof of Theorem 4.4 is presented in Section B.6 of the supplementary, and the proof consists of three steps. In step 1, it is proved that there exists a finite $k_0 \geq k_1$ such that for all $k > k_0$,

$$F(\mathbf{x}^{(k)}) - F_{\mathbf{m}^*}(\bar{\mathbf{x}}) \leq \mathcal{O}(\frac{1}{k^2}).$$

Noting that $F(\mathbf{x}^{(k)}) = F_{\mathbf{m}^*}(\mathbf{x}^{(k)}) \geq F_{\mathbf{m}^*}(\bar{\mathbf{x}})$ by the optimality of $\bar{\mathbf{x}}$, the above inequality combined with the monotone nonincreasing of $\left\{ F(\mathbf{x}^{(k)}) \right\}$ indicate that $F(\mathbf{x}^{(k)}) \downarrow F_{\mathbf{m}^*}(\bar{\mathbf{x}})$.

In step 2, it is proved that $F(\mathbf{x}') = F_{\mathbf{m}^*}(\bar{\mathbf{x}})$ for any limit point $\mathbf{x}' \in \Omega$. According to the definition of $F_{\mathbf{m}^*}$ and the optimality of $\bar{\mathbf{x}}$, it follows that $F(\mathbf{x}')$ is a local minimum of $F$ and a global minimum of $F_{\mathbf{m}^*}$ over $\mathcal{R}^*$.

In step 3, it is proved that any limit point $\mathbf{x}' \in \Omega$ is a critical point of $F$ under a mild condition, following the argument in step 2.

## 5 EXPERIMENTAL RESULTS

In this section, PPGD and the baseline optimization methods are used to solve the capped-$\ell^1$ regularized logistic regression problem, $\min_{\mathbf{x}} \frac{1}{n} \sum_{i=1}^{n} \log(1 + \exp(-y_i \mathbf{x}^\top x_i)) + h(\mathbf{x})$, where $\{x_i, y_i\}_{i=1}^{n}$ are training data, $g(\mathbf{x}) = \frac{1}{n} \sum_{i=1}^{n} \log(1 + \exp(-y_i \mathbf{x}^\top x_i))$, and $h(\mathbf{x}) = \sum_{i=1}^{d} f(\mathbf{x}_i)$ with $f(x) = \lambda \min \{|x|, b\}$. We set $\lambda = 0.2$ and conduct experiments on the MNIST handwritten digits dataset (LeCun et al., 1998).

The MNIST data set contains $70,000$ examples of handwritten digit images of $10$ classes. Each image is of size $28 \times 28$ and represented by a 784-dimensional vector. The CIFAR-10 dataset contains 60000 images of size $32 \times 32$ with 10 classes. For each dataset, we randomly sample two classes with 5000 images in each class, which form the training data $\{\mathbf{x}_i, y_i\}_{i=1}^{10000}$ with $\{\mathbf{x}_i\}_{i=1}^{10000}$ being the sampled images and $\{y_i\}_{i=1}^{10000}$ being the corresponding class labels ($\pm 1$). We compare PPGD to the regular monotone Accelerated Proximal Gradient (APG) described in Algorithm 4 and the monotone Accelerated Proximal Gradient (mAPG) algorithm (Li & Lin, 2015). Due to the observation that monotone version of accelerated gradient methods usually converge faster than its nonmonotone counterparts, the regular nonmonotone APG and the nonmonotone Accelerated Proximal Gradient (nmAPG) method (Li & Lin, 2015) are not included in the baselines.

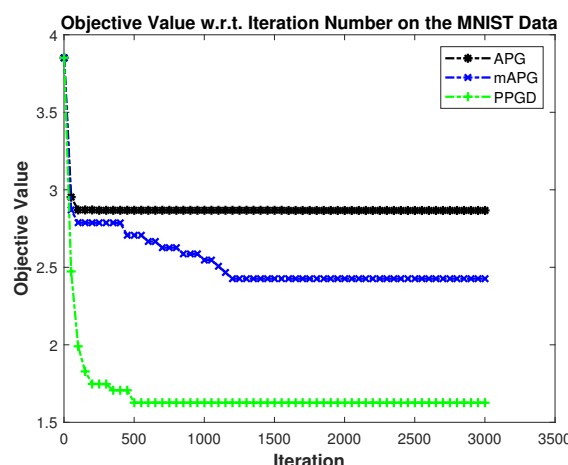

Figure 3: Illustration of the objective value with respect to the iteration number on the MNIST data for capped-$\ell^1$ regularized logistic regression

Figure 3 illustrate the decrease of objective value with respect to the iteration number on the MNIST dataset. It can be observed that PPGD always converges faster than all the competing baselines, evidencing the proved fast convergence results in this paper.

## 6 CONCLUSION

We present Projective Proximal Gradient Descent (PPGD) to solve a class of challenging nonconvex and nonsmooth optimization problems. Using a novel projection operator and a carefully designed Negative-Curvature-Exploitation algorithm, PPGD locally matches Nesterov's optimal convergence rate of first-order methods on smooth and convex objective function. The effectiveness of PPGD is evidenced by experiments.

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

## A   MORE DETAILS ABOUT THIS PAPER

### A.1   KURDYKA-ŁOJASIEWICZ (KŁ) PROPERTY

**Definition A.1** (Kurdyka-Łojasiewicz (KŁ) Property). Let $u\colon \mathbb{R}^d \to (-\infty, \infty]$ be proper and lower semicontinuous. $u$ is said to satisfy the Kurdyka-Lojasiewicz (KŁ) property at $\mathbf{u} \in \operatorname{dom} \partial u :=$ $\{\mathbf{u}' \in \mathbb{R}^d\colon \partial u(\mathbf{u}') \neq \emptyset\}$, if there exist $\eta \in (0, \infty]$, a neighborhood $U$ of $\mathbf{u}$ and a function $\phi \in \Phi_\eta$, such that for all $\bar{\mathbf{u}} \in U \cup [u(\mathbf{u}) < u(\bar{\mathbf{u}}) < u(\mathbf{u}) + \eta]$, the following inequality holds:

$$\phi'\big(u(\bar{\mathbf{u}}) - u(\mathbf{u})\big)\operatorname{dist}(\mathbf{0}, \partial u(\bar{\mathbf{u}})) \geq 1.$$

Here $\eta \in (0, \infty]$, $\Phi_\eta$ is defined to be the class of all concave and continuous functions $\phi\colon [0, \eta) \to \mathbb{R}^+$ which satisfy the following conditions:

1. $\phi(0) = 0$;

2. $\phi$ is continuously differentiable on $(0, \eta)$ and continuous at $0$;

3. $\phi'(r) > 0$ for all $r \in (0, \eta)$.

If $u$ satisfies the KŁ property at each point of $\operatorname{dom} \partial u$, then $u$ is called a KŁ function.

### A.2   PROXIMAL GRADIENT DESCENT (PGD) AND MONOTONE ACCELERATED PROXIMAL GRADIENT (APG) DESCENT

We describe PGD and the regular monotone APG in this subsection.

In the $k$-th iteration of PGD for $k \geq 1$, gradient descent is performed on the squared loss term $g(\mathbf{x})$ to obtain an intermediate variable as the result of gradient descent, i.e. $\mathbf{x}^{(k)} - s\nabla g(\mathbf{x}^{(k)})$, where $s > 0$ is the step size, and $\frac{1}{s}$ is usually chosen to be larger than $L_g$. $\mathbf{x}^{(k+1)}$ is computed by proximal mapping on $\mathbf{x}^{(k)} - s\nabla g(\mathbf{x}^{(k)})$.

---

**Algorithm 3** Proximal Gradient Descent $(\mathbf{x}^{(0)})$

---

1: **for** k=1,..., **do**

$$\mathbf{x}^{(k+1)} = \mathsf{prox}_{sh}\left(\mathbf{x}^{(k)} - s\nabla g(\mathbf{x}^{(k)})\right) \tag{18}$$

---

**Algorithm 4** Monotone Accelerated Proximal Gradient Descent $(\mathbf{x}^{(0)})$

---

1: $\mathbf{z}^{(1)} = \mathbf{x}^{(1)} = \mathbf{x}^{(0)}$, $t_0 = 0$.
2: **for** $k = 1, \ldots,$ **do**

$$\mathbf{u}^{(k)} = \mathbf{x}^{(k)} + \frac{t_{k-1}}{t_k}(\mathbf{z}^{(k)} - \mathbf{x}^{(k)}) + \frac{t_{k-1} - 1}{t_k}(\mathbf{x}^{(k)} - \mathbf{x}^{(k-1)}),$$
$$\mathbf{z}^{(k+1)} = \mathsf{prox}_{sh}(\mathbf{u}^{(k)} - s\nabla g(\mathbf{u}^{(k)})),$$
$$t_{k+1} = \frac{\sqrt{1 + 4t_k^2} + 1}{2},$$
$$\mathbf{x}^{(k+1)} = \begin{cases} \mathbf{z}^{(k+1)} & \text{if } F(\mathbf{z}^{(k+1)}) \leq F(\mathbf{x}^{(k)}) \\ \mathbf{x}^{(k)} & \text{otherwise.} \end{cases}$$

---

The iterations start from $k = 1$ and continue until the sequence $\{F(\mathbf{x}^{(k)})\}_k$ or $\{\mathbf{x}^{(k)}\}_k$ converges or maximum iteration number is achieved. The optimization algorithm for the $\ell^0$ sparse approximation problem (1) by PGD is described in Algorithm 3. In practice, the time complexity of optimization by PGD is $\mathcal{O}(Mdn)$ where $M$ is the number of iterations (or maximum number of iterations) for PGD.

The regular monotone Accelerated Proximal Gradient (APG) descent algorithm is described in Algorithm 4.

## B    PROOFS

The following lemma shows that the Fermat's rule still applies to functions in the sense of Fréchet subdifferential, that is, local minimum has $\mathbf{0}$ in its Fréchet subdifferential and also its limiting subdifferential.

**Lemma B.1.** (Fermat's rule for Fréchet subdifferential and limiting subdifferential, also in Rockafellar & Wets (2009, Theorem 10.1)) Let $u$ be a real-valued function defined on $\Omega \subseteq \mathbb{R}^d$, $u \coloneqq \Omega \to \mathbb{R}$, has a local minimum point at $\mathbf{x} \in \Omega$. Then $\mathbf{0} \in \tilde{\partial} u(\mathbf{x}) \subseteq \partial u(\mathbf{x})$.

*Proof.* Since $\mathbf{x} \in \Omega$ is a local minimum point, then there exists a neighborhood $\mathbf{x} \in \mathcal{U}(\mathbf{x})$ such that $u(\mathbf{y}) \geq u(\mathbf{x})$ for all $\mathbf{y} \in \mathcal{U}(\mathbf{x})$. It follows that

$$\liminf_{\mathbf{y} \neq \mathbf{x}, \mathbf{y} \to \mathbf{x}, \mathbf{y} \in \mathbf{dom} u} \frac{u(\mathbf{y}) - u(\mathbf{x})}{\|\mathbf{y} - \mathbf{x}\|} \geq 0. \tag{19}$$

By (19), $\mathbf{0} \in \tilde{\partial} u(\mathbf{x})$. Because $\tilde{\partial} u(\mathbf{x}) \subseteq \partial u(\mathbf{x})$, $\mathbf{0} \in \tilde{\partial} u(\mathbf{x}) \subseteq \partial u(\mathbf{x})$. $\qquad \square$

We need to have the following two lemmas before proving Lemma 4.1.

**Lemma B.2** (Decrease of $f$ at continuous endpoint)**.** Suppose Assumption 1 and Assumption 2 hold, $s < \min \left\{ \frac{s_0}{G+F_0}, \frac{\varepsilon_0}{L_g(1-w_0)(G+F_0)} \right\}$, $\left\| \nabla g(\mathbf{w}^{(k)}) \right\|_2 \leq G$, $P(\mathbf{x}_i^{(k+1)}) \neq P(\mathbf{x}_i^{(k)}) = m_i$ for some $k \geq 0$ and some $i \in [d]$, and $f$ is continuous at $q(\mathbf{w}_i^{(k)})$. Then $f(\mathbf{x}_i^{(k+1)}) \leq f_{m_i}(\mathbf{z}_i^{(k+1)})$. Furthermore, if $d_{i,1} \geq w_0 d_{i,0}$, then

$$f(\mathbf{x}_i^{(k+1)}) \leq f_{m_i}(\mathbf{z}_i^{(k+1)}) - \kappa_1, \tag{20}$$

where $\kappa_1 \coloneqq sCw_0 (\varepsilon_0 - sL_g(1-w_0)(G+F_0))$.

**Lemma B.3** (Decrease of $f$ at discontinuous endpoints)**.** Suppose Assumption 1 holds, $s < \min \left\{ \frac{s_0}{G+F_0}, \frac{J}{2F_0(G+F_0)} \right\}$, $\left\| \nabla g(\mathbf{w}^{(k)}) \right\|_2 \leq G$, $P(\mathbf{x}_i^{(k+1)}) \neq P(\mathbf{x}_i^{(k)}) = m_i$ for some $k \geq 0$ and some $i \in [d]$. Then

$$f(\mathbf{x}_i^{(k+1)}) \leq f_{m_i}(\mathbf{z}_i^{(k+1)}) - \kappa_2, \tag{21}$$

where $\kappa_2 \coloneqq J - 2sF_0(G+F_0)$.

### B.1    PROOF OF LEMMA B.2

**Proof of Lemma B.2.** Let $q = q(\mathbf{w}_i^{(k)})$. We must have $\mathbf{x}_i^{(k+1)} = \mathbf{z}_i^{(k+1)}$ with the PPGD algorithm and the Negative-Curvature-Exploitation algorithm described in Algorithm 1 and Algorithm 2. Without loss of generality, we assume $\mathbf{x}_i^{(k+1)} \in \mathcal{R}_{m_i}^+$. The case that $\mathbf{x}_i^{(k+1)} \in \mathcal{R}_{m_i}^-$ can be proved in a similar manner.

We have

$$f_{m_i}(\mathbf{x}_i^{(k+1)}) = f(q) + v^- \left( \mathbf{x}_i^{(k+1)} - q \right) \tag{22}$$

by definition (5), where $v^- = \lim_{x \to q^-} f'(x)$. On the other hand, we have

$$\mathbf{z}_i^{(k+1)} - \mathbf{w}_i^{(k)} = -s \left( \left[ \nabla g(\mathbf{w}^{(k)}) \right]_i + f'_{m_i}(\mathbf{z}_i^{(k+1)}) \right) = -s \left( \left[ \nabla g(\mathbf{w}^{(k)}) \right]_i + v^- \right). \tag{23}$$

It follows by (23) that

$$\left| \mathbf{x}_i^{(k+1)} - q \right| = \left| \mathbf{z}_i^{(k+1)} - q \right| \leq \left| \mathbf{z}_i^{(k+1)} - \mathbf{w}_i^{(k)} \right| = s \left| \left[ \nabla g(\mathbf{w}^{(k)}) \right]_i + v^- \right| \leq s(G+F_0). \tag{24}$$

As a result, $s < \frac{s_0}{G+F_0}$ guarantees that $\mathbf{x}_i^{(k+1)} \in (q, q+s_0)$. Therefore, we have

$$f(\mathbf{x}_i^{(k+1)}) \leq f(x) + f'(\mathbf{x}_i^{(k+1)}) \left( \mathbf{x}_i^{(k+1)} - x \right) \tag{25}$$

when $x \in (q, \mathbf{x}_i^{(k+1)})$. Letting $x \to q$ in (25), we have

$$f(\mathbf{x}_i^{(k+1)}) \leq f(q) + v^+ \left( \mathbf{x}_i^{(k+1)} - q \right), \tag{26}$$

where $v^+ = \lim_{x \to q^+} f'(x)$. By the negative curvature condition in Assumption 1(d), we have $v^- - v^+ \geq C > 0$, therefore,

$$f(\mathbf{x}_i^{(k+1)}) \leq f_{m_i}(\mathbf{x}_i^{(k+1)}) - C\left(\mathbf{x}_i^{(k+1)} - q\right)$$

$$\leq f_{m_i}(\mathbf{x}_i^{(k+1)}) - Cw_0\left(\mathbf{z}_i^{(k+1)} - \mathbf{w}_i^{(k)}\right). \tag{27}$$

Because $\mathbf{z}_i^{(k+1)} \in \mathcal{R}_{m_i}^+$, $\mathbf{z}_i^{(k+1)} \geq \mathbf{w}_i^{(k)}$. By (27) we have

$$f(\mathbf{x}_i^{(k+1)}) \leq f_{m_i}(\mathbf{x}_i^{(k+1)}) = f_{m_i}(\mathbf{z}_i^{(k+1)}). \tag{28}$$

Let $\tilde{\mathbf{w}} \in \mathbb{R}^d$ with $\tilde{\mathbf{w}}_j = \mathbf{w}_j^{(k)}$ for all $j \neq i$, and $\tilde{\mathbf{w}}_i = q$. By the definition of $\tilde{\mathbf{w}}$ and (23), we have

$$\left\|\mathbf{w}^{(k)} - \tilde{\mathbf{w}}\right\|_2 = \left|\mathbf{w}_i^{(k)} - q\right| \leq (1 - w_0)\left|\mathbf{z}_i^{(k+1)} - \mathbf{w}_i^{(k)}\right|$$

$$= s(1 - w_0)\left|\left[\nabla g(\mathbf{w}^{(k)})\right]_i + v^-\right| \leq s(1 - w_0)(G + F_0). \tag{29}$$

We have

$$\mathbf{z}_i^{(k+1)} - \mathbf{w}_i^{(k)} = \left|\mathbf{z}_i^{(k+1)} - \mathbf{w}_i^{(k)}\right| = s\left|[\nabla g(\mathbf{w}^{(k)})]_i + v^-\right|$$

$$= s\left|\left[\nabla g(\mathbf{w}^{(k)}) - \nabla g(\tilde{\mathbf{w}})\right]_i + [\nabla g(\tilde{\mathbf{w}})]_i + v^-\right|$$

$$\geq s\left(\varepsilon_0 - L_g\left\|\mathbf{w}^{(k)} - \tilde{\mathbf{w}}\right\|_2\right)$$

$$\geq s\left(\varepsilon_0 - sL_g(1 - w_0)(G + F_0)\right), \tag{30}$$

where the last inequality follows from (29).

It follows by (27) and (30) that

$$f(\mathbf{x}_i^{(k+1)}) \leq f_{m_i}(\mathbf{x}_i^{(k+1)}) - sCw_0\left(\varepsilon_0 - sL_g(1 - w_0)(G + F_0)\right). \tag{31}$$

Noting that $\mathbf{x}_i^{(k+1)} = \mathbf{z}_i^{(k+1)}$ in (31) completes the proof. $\qquad\square$

## B.2 Proof of Lemma B.3

Before presenting the proof, we introduce and prove the following lemma.

**Lemma B.4.** Suppose Assumption 1 holds, $\left\|\mathbf{w}^{(k)}\right\|_2 \leq G$, and $s < \min\left\{\frac{J}{F_0G + G^2/2}, \frac{s_0}{G}\right\}$, $P(\mathbf{x}_i^{(k+1)}) \neq P(\mathbf{x}_i^{(k)}) = m_i$ for some $k \geq 0$ and some $i \in [d]$, and $f$ is not continuous at $q(\mathbf{w}_i^{(k)})$. Then $f(q(\mathbf{w}_i^{(k)})) < \lim_{y \to q(\mathbf{w}_i^{(k)})^-} f(y)$ if $q(\mathbf{w}_i^{(k)})$ is the right endpoint of $\mathcal{R}_{m_i}$ for $1 \leq m_i < M$, and $f(q(\mathbf{w}_i^{(k)})) < \lim_{y \to q(\mathbf{w}_i^{(k)})^+} f(y)$ if $q(\mathbf{w}_i^{(k)})$ is the left endpoint of $\mathcal{R}_{m_i}$ for $1 < m_i \leq M$.

**Proof of Lemma B.4.** We prove the case when $q(\mathbf{w}_i^{(k)})$ is the right endpoint of $\mathcal{R}_{m_i}$. The case when $q(\mathbf{w}_i^{(k)})$ is the left endpoint of $\mathcal{R}_{m_i}$ can be proved in a similar manner.

Suppose the claimed result $f(q(\mathbf{w}_i^{(k)})) < \lim_{y \to q(\mathbf{w}_i^{(k)})^-} f(y)$ does not hold, so we must have

$\lim_{y \to q(\mathbf{w}_i^{(k)})^+} f(y) > f(q(\mathbf{w}_i^{(k)}))$. In this case, $\mathbf{x}_i^{(k+1)} = \mathbf{z}_i^{(k+1)}$, and $\mathbf{z}_i^{(k+1)} \in (q(\mathbf{w}_i^{(k)}), \infty)$.

Define $p(v) := \frac{1}{2s}\left(v - \left[\mathbf{w}^{(k)} - s\nabla g(\mathbf{w}^{(k)})\right]_i\right)^2 + f_{m_i}(v)$. We have

$$\mathbf{z}_i^{(k+1)} = \mathsf{prox}_{sf_{P(\mathbf{x}_i^{(k)})}}\left(\left[\mathbf{w}^{(k)} - s\nabla g(\mathbf{w}^{(k)})\right]_i\right).$$

The optimality of $\mathbf{z}_i^{(k+1)}$ by Fermat's rule indicates that $\mathbf{z}_i^{(k+1)} = \left[\mathbf{w}^{(k)} - s\nabla g(\mathbf{w}^{(k)})\right]_i$, and

$$p(\mathbf{z}_i^{(k+1)}) \leq p(\mathbf{w}_i^{(k)}). \tag{32}$$

We have $\mathbf{w}_i^{(k)} \in (q(\mathbf{w}_i^{(k)}) - s_0, q(\mathbf{w}_i^{(k)}))$ due to $s < \frac{s_0}{G}$.

On the other hand, we have

$$p(\mathbf{w}_i^{(k)}) = \frac{s}{2}\nabla\left[g(\mathbf{w}^{(k)})\right]_i^2 + f_{m_i}(\mathbf{w}_i^{(k)}), \quad p(\mathbf{z}_i^{(k+1)}) \geq f_{m_i}(\mathbf{z}_i^{(k+1)}) = \lim_{y \to q(\mathbf{w}_i^{(k)})+} f(y), \quad (33)$$

and

$$\left|f_{m_i}(\mathbf{w}_i^{(k)}) - f_{m_i}(q(\mathbf{w}_i^{(k)}))\right| = \left|f(\mathbf{w}_i^{(k)}) - f(q(\mathbf{w}_i^{(k)}))\right| \leq F\left|\mathbf{w}_i^{(k)} - q(\mathbf{w}_i^{(k)})\right|$$
$$\leq F\left|\mathbf{w}_i^{(k)} - \mathbf{z}_i^{(k+1)}\right| \leq sF_0\left[\nabla g(\mathbf{w}^{(k)})\right]_i \leq sF_0 G. \quad (34)$$

(34) follows because .

It follows by (33) and (34) that

$$p(\mathbf{z}_i^{(k+1)}) \geq f(q(\mathbf{w}_i^{(k)})) + J \overset{\textcircled{1}}{>} f_{m_i}(q(\mathbf{w}_i^{(k)})) + sF_0 G + \frac{s}{2}G^2$$
$$\geq f_{m_i}(\mathbf{w}_i^{(k)}) + \frac{s}{2}G^2$$
$$\geq p(\mathbf{w}_i^{(k)}), \quad (35)$$

where $\textcircled{1}$ follows from $s < \frac{J}{F_0 G + G^2/2}$. This contradiction shows that we must have $\lim_{y \to q(\mathbf{w}_i^{(k)})+} f(y) \leq f(q(\mathbf{w}_i^{(k)}))$. Since $f$ is not continuous at $q(\mathbf{w}_i^{(k)})$, we must have $f(q(\mathbf{w}_i^{(k)})) < \lim_{y \to q(\mathbf{w}_i^{(k)})-} f(y)$. $\qquad\square$

**Proof of Lemma B.3**. According to Lemma B.4, the following claim holds: when $q(\mathbf{w}_i^{(k)})$ is a right endpoint of $\mathcal{R}_{m_i}$, then $f(q(\mathbf{w}_i^{(k)})) < \lim_{y \to q(\mathbf{w}_i^{(k)})-} f(y)$; when $q(\mathbf{w}_i^{(k)})$ is a left endpoint of $\mathcal{R}_{m_i}$, then $f(q(\mathbf{w}_i^{(k)})) < \lim_{y \to q(\mathbf{w}_i^{(k)})+} f(y)$.

Let $q = q(\mathbf{w}_i^{(k)})$. Without loss of generality, we assume $\mathbf{x}_i^{(k+1)} \in \mathcal{R}_{m_i}^+$. The case that $\mathbf{x}_i^{(k+1)} \in \mathcal{R}_{m_i}^-$ can be proved in a similar manner.

We first consider the case that $\mathcal{R}_{m_i+1}$ is not a single-point set, that is, $\mathcal{R}_{m_i+1} \neq \{q\}$.

According to the definition of surrogate function (5), we have

$$f_{m_i}(\mathbf{x}_i^{(k+1)}) = \lim_{y \to q^-} f(y) + v^-(\mathbf{x}_i^{(k+1)} - q), \quad (36)$$

where $v^- = \lim_{x \to q^-} f'(x)$. If $\mathbf{x}_i^{(k+1)} \neq q$, applying the argument in (24) in the proof of Lemma B.2, $s < \frac{s_0}{G+F_0}$ guarantees that $\mathbf{x}_i^{(k+1)} \in (q, q + s_0)$. So we have

$$f(\mathbf{x}_i^{(k+1)}) \leq f(x) + f'(\mathbf{x}_i^{(k+1)})\left(\mathbf{x}_i^{(k+1)} - x\right) \quad (37)$$

when $x \in (q, \mathbf{x}_i^{(k+1)})$. Letting $x \to q$ in (37), we have

$$f(\mathbf{x}_i^{(k+1)}) \leq f(q) + v^+\left(\mathbf{x}_i^{(k+1)} - q\right), \quad (38)$$

where $v^+ = \lim_{x \to q^+} f'(x)$. In addition,

It follows by (38) and (24) in the proof of Lemma B.2 that

$$f(\mathbf{x}_i^{(k+1)}) \leq f(q) + sF_0(G + F_0). \quad (39)$$

We note that (39) holds for all $\mathbf{x}_i^{(k+1)} \in [q, q + s_0)$ Moreover, it follows by (36) and (24) in the proof of Lemma B.2 that

$$f_{m_i}(\mathbf{x}_i^{(k+1)}) \geq \lim_{y \to q^-} f(y) - \left|v^-(\mathbf{x}_i^{(k+1)} - q)\right| \geq \lim_{y \to q^-} f(y) - sF_0(G + F_0) \geq f(q) + J - sF_0(G + F_0).$$
$$(40)$$

Combining (39) and (40), we have

$$f(\mathbf{x}_i^{(k+1)}) \le f_{m_i}(\mathbf{x}_i^{(k+1)}) - (J - 2sF_0(G + F_0)) = f_{m_i}(\mathbf{z}_i^{(k+1)}) - (J - 2sF_0(G + F_0)). \tag{41}$$

If $\mathcal{R}_{P(\mathbf{z}_i^{(k+1)})} = \{q\}$, then $\mathbf{x}_i^{(k+1)} = q$ and $f(\mathbf{x}_i^{(k+1)}) = f(q)$. By the fact that

$$f_{m_i}(\mathbf{z}_i^{(k+1)}) = \lim_{y \to q^-} f(y) + v^-(\mathbf{z}_i^{(k+1)} - q)$$

and $\left|\mathbf{z}_i^{(k+1)} - q\right| \le \left|\mathbf{z}_i^{(k+1)} - \mathbf{w}_i^{(k)}\right|$, following the argument similar to (40) we have

$$f_{m_i}(\mathbf{z}_i^{(k+1)}) \ge f(q) + J - sF_0(G + F_0) = f(\mathbf{x}_i^{(k+1)}) + J - sF_0(G + F_0).$$

It follows that

$$f(\mathbf{x}_i^{(k+1)}) \le f_{m_i}(\mathbf{z}_i^{(k+1)}) - (J - sF_0(G + F_0)). \tag{42}$$

The proof is completed by combining (41) and (42).

$\square$

## B.3 PROOF OF LEMMA 4.1

**Proof of Lemma 4.1.** We split $[d]$ into three disjoint subsets, $[d] = S_1 \cup S_2 \cup S_3$, and the three subsets are defined by

$$S_1 := \left\{ i \in [d] \colon P(\mathbf{x}_i^{(k+1)}) \ne P(\mathbf{x}_i^{(k)}), f \text{ is continuous at } q(\mathbf{w}_i^{(k)}) \right\}$$

$$S_2 := \left\{ i \in [d] \colon P(\mathbf{x}_i^{(k+1)}) \ne P(\mathbf{x}_i^{(k)}), f \text{ is not continuous at } q(\mathbf{w}_i^{(k)}) \right\},$$

$$S_3 := \left\{ i \in [d] \colon P(\mathbf{x}_i^{(k+1)}) = P(\mathbf{x}_i^{(k)}) \right\}. \tag{43}$$

Let $m_i = P(\mathbf{x}_i^{(k)})$. According to Lemma B.2, for all $i \in S_1$ such that $d_{i,1} \ge w_0 d_{i,0}$, we have

$$f(\mathbf{x}_i^{(k+1)}) \le f_{m_i}(\mathbf{z}_i^{(k+1)}) - \kappa_1. \tag{44}$$

In addition, for all $i \in S_1$, we have

$$f(\mathbf{x}_i^{(k+1)}) \le f_{m_i}(\mathbf{z}_i^{(k+1)}). \tag{45}$$

According to Lemma B.3,

$$f(\mathbf{x}_i^{(k+1)}) \le f_{m_i}(\mathbf{z}_i^{(k+1)}) - \kappa_2. \tag{46}$$

for all $i \in S_2$.

For all $i \in S_3$, we have

$$f(\mathbf{x}_i^{(k+1)}) = f_{m_i}(\mathbf{x}_i^{(k+1)}) = f_{m_i}(\mathbf{z}_i^{(k+1)}). \tag{47}$$

The Negative-Curvature-Exploitation algorithm described in Algorithm 2 guarantees that when $P(\mathbf{x}^{(k+1)}) \ne P(\mathbf{x}^{(k)})$, there exists at lease one $i \in S_1 \cup S_2$ such that (44) or (46) holds. It follows by (44)-(47) and the above argument that

$$\sum_{i=1}^d f(\mathbf{x}_i^{(k+1)}) \le \sum_{i=1}^d f_{m_i}(\mathbf{z}_i^{(k+1)}) - \kappa_0. \tag{48}$$

Let $\bar{S}_{k+1} := \left\{ i \in [d] \colon \mathbf{x}_i^{(k+1)} \ne \mathbf{z}_i^{(k+1)} \right\}$. It can be verified by Algorithm 2 that $\bar{S}_{k+1} \subseteq S_2$. If $\bar{S}_{k+1} = \emptyset$, then $\mathbf{x}^{(k+1)} = \mathbf{z}^{(k+1)}$. It follows from (48) that

$$F(\mathbf{x}^{(k+1)}) \le g(\mathbf{x}^{(k+1)}) + \sum_{i=1}^d f_{m_i}(\mathbf{z}_i^{(k+1)}) - \kappa = F_{P(\mathbf{x}^{(k)})}(\mathbf{z}^{(k+1)}) - \kappa_0. \tag{49}$$

If $\bar{S}_{k+1} \neq \emptyset$, then it is possible that $\mathbf{x}^{(k+1)} \neq \mathbf{z}^{(k+1)}$. To handle the case that $\mathbf{x}^{(k+1)} \neq \mathbf{z}^{(k+1)}$, we first bound $\left\| \mathbf{x}^{(k+1)} - \mathbf{z}^{(k+1)} \right\|_2$. Define $h_{\mathbf{m}}(\mathbf{x}) := \sum_{i=1}^{d} f_{\mathbf{m}_i}(\mathbf{x}_i)$ for $\mathbf{m} \in \mathbb{N}^d$ and $\mathbf{m}_i \in m_i = P(\mathbf{x}_i^{(k)})$ for all $i \in [d]$. By the optimality of $\mathbf{z}^{(k+1)}$, we have

$$\frac{1}{s}(\mathbf{w}^{(k)} - \mathbf{z}^{(k+1)}) - \nabla g(\mathbf{w}^k) \in \tilde{\partial} h_{\mathbf{m}}(\mathbf{w}^k). \tag{50}$$

It follows from (50) that $\left\| \mathbf{w}^{(k)} - \mathbf{z}^{(k+1)} \right\|_2 \leq s(G + \sqrt{d}F_0)$, and $\left\| \mathbf{x}^{(k+1)} - \mathbf{z}^{(k+1)} \right\|_2 \leq \left\| \mathbf{w}^{(k)} - \mathbf{z}^{k+1} \right\|_2 \leq s(G + \sqrt{d}F_0)$. We then bound $\left| g(\mathbf{x}^{(k+1)}) - g(\mathbf{z}^{(k+1)}) \right|$ by

$$\left| g(\mathbf{x}^{(k+1)}) - g(\mathbf{z}^{(k+1)}) \right| \overset{\text{①}}{=} \left| \left\langle \nabla g(\zeta), \mathbf{x}^{(k+1)} - \mathbf{z}^{(k+1)} \right\rangle \right| \leq \left\| \nabla g(\zeta) \right\|_2 \left\| \mathbf{x}^{(k+1)} - \mathbf{z}^{(k+1)} \right\|_2$$

$$\leq \left\| \nabla g(\zeta) - \nabla g(\mathbf{w}^{(k)}) + \nabla g(\mathbf{w}^{(k)}) \right\|_2 \cdot s(G + \sqrt{d}F_0)$$

$$\overset{\text{②}}{\leq} \left( L_g \left\| \zeta - \mathbf{w}^{(k)} \right\|_2 + G \right) \cdot s(G + \sqrt{d}F_0)$$

$$\leq s \left( sL_g(G + \sqrt{d}F_0) + G \right) (G + \sqrt{d}F_0). \tag{51}$$

Here $\zeta$ in ① lies in the line segment between $\mathbf{x}^{(k+1)}$ and $\mathbf{z}^{(k+1)}$ by the mean value theorem of differentiable functions. ② follows from the fact that $\left\| \zeta - \mathbf{w}^{(k)} \right\|_2 \leq \left\| \mathbf{w}^{(k)} - \mathbf{z}^{(k+1)} \right\|_2 \leq s(G + \sqrt{d}F_0)$. We then have

$$F(\mathbf{x}^{(k+1)}) \leq g(\mathbf{x}^{(k+1)}) + \sum_{i=1}^{d} f_{m_i}(\mathbf{z}_i^{(k+1)}) - \kappa_0$$

$$\leq g(\mathbf{z}^{(k+1)}) + \sum_{i=1}^{d} f_{m_i}(\mathbf{z}_i^{(k+1)}) + \left( g(\mathbf{x}^{(k+1)}) - g(\mathbf{z}^{(k+1)}) \right) - \kappa_0$$

$$\leq F_{P(\mathbf{x}^{(k)})}(\mathbf{z}^{(k+1)}) - \left( \kappa_0 - s \left( sL_g(G + \sqrt{d}F_0) + G \right) (G + \sqrt{d}F_0) \right)$$

$$= F_{P(\mathbf{x}^{(k)})}(\mathbf{z}^{(k+1)}) - \kappa. \tag{52}$$

The PPGD algorithm guarantees that

$$F_{P(\mathbf{x}^{(k)})}(\mathbf{z}^{(k+1)}) \leq F(\mathbf{x}^{(k)}). \tag{53}$$

It follows from (49), (52), and (53) that

$$F(\mathbf{x}^{(k+1)}) \leq F(\mathbf{x}^{(k)}) - \left( \kappa_0 - s \left( sL_g(G + \sqrt{d}F_0) + G \right) (G + \sqrt{d}F_0) \right).$$

Since $s < \dfrac{-G + \sqrt{G^2 + \frac{4A\kappa_0}{G + \sqrt{d}F_0}}}{2A}$, we have $\kappa > 0$, which completes the proof. $\qquad \square$

### B.4 PROOF OF LEMMA 4.2

**Proof of Lemma 4.2.** The PPGD algorithm described in Algorithm 1 ensures that $F(\mathbf{x}^{(k)}) \leq F(\mathbf{x}^{(k-1)})$ for $k = 1$, and $\left\| \nabla g(\mathbf{w}^{(k)}) \right\|_2 \leq G$ for $k = 1$.

Suppose that $F(\mathbf{x}^{(k)}) \leq F(\mathbf{x}^{(k-1)})$ and $\left\| \nabla g(\mathbf{w}^{(k)}) \right\|_2 \leq G$ hold for all $1 \leq k \leq k'$ with $k' \geq 1$. With the chosen step size and the proof of Lemma 4.1, we have $F(\mathbf{x}^{(k'+1)}) \leq F(\mathbf{x}^{(k')})$. This indicates that $\mathbf{x}^{(k'+1)} \in \mathcal{L}$ and $\mathbf{w}^{(k'+1)} \in \mathcal{L}_{R_0}$, so $\left\| \mathbf{w}^{(k'+1)} \right\|_2 \leq G$. It follows by induction that $F(\mathbf{x}^{(k)}) \leq F(\mathbf{x}^{(k-1)})$ and $\left\| \nabla g(\mathbf{w}^{(k)}) \right\|_2 \leq G$ hold for all $k \geq 1$. $\qquad \square$

### B.5 PROOF OF THEOREM 4.3

**Proof of Theorem 4.3.** By Lemma 4.2, $\left\| \nabla g(\mathbf{w}^{(k)}) \right\|_2 \leq G$ holds for all $k \geq 1$. Then the conclusion of this theorem directly follows from Lemma 4.1.

$\square$

### B.6 PROOF OF THEOREM 4.4

The following lemma is crucial in the proof of Theorem 4.4. It shows that after sufficient iterations, all the coordinates of $\mathbf{x}^{(k)}$ belong to the same the convex pieces indexed by $\mathbf{m}^*$, that is, $P(\mathbf{x}^{(k)}) = \mathbf{m}^*$. Moreover, the objective value $F(\mathbf{x}^{(k)})$ is not greater than the surrogate objective value $F_{\mathbf{m}^*}(\mathbf{z}^{(k)}) = g(\mathbf{z}^{(k)}) + \sum_{i=1}^{d} f_{\mathbf{m}_i^*}(\mathbf{z}_i^{(k)})$.

**Lemma B.5.** Suppose Assumption 1 and Assumption 2 hold, and $s < \min\left\{s_1, \frac{\varepsilon_0}{L_g(G+\sqrt{d}F_0)}\right\}$. Then there must exists a finite $\bar{k} \in \mathbb{N}$ such that $P(\mathbf{x}^{(k)}) = \mathbf{m}^* \in \mathbb{N}^d$ and $F(\mathbf{x}^{(k)}) \leq F_{\mathbf{m}^*}(\mathbf{z}^{(k)})$ for all $k \geq \bar{k}$.

**Proof of Lemma B.5.** According to Theorem 4.3, when $P(\mathbf{x}^{(k+1)}) \neq P(\mathbf{x}^{(k)})$, $F(\mathbf{x}^{(k+1)}) \leq F(\mathbf{x}^{(k)}) - \kappa$. Because $\inf_{x \in \mathbb{R}^d} F(x) > -\infty$, we can only have finite number of $k$'s such that $F(\mathbf{x}^{(k+1)}) \leq F(\mathbf{x}^{(k)}) - \kappa$. As a result, there must exists a finite $k_1 \in \mathbb{N}$ such that $P(\mathbf{x}^{(k)}) = \mathbf{m}^* \in \mathbb{N}^d$ for all $k \geq k_1$.

We now prove that there exists a finite $k_2 > k_1$ such that $F(\mathbf{x}^{(k)}) \leq F_{\mathbf{m}^*}(\mathbf{z}^{(k)})$ for all $k \geq k_2$. Suppose this is not the case and there are infinitely many $k$'s such that $F(\mathbf{x}^{(k)}) > F_{\mathbf{m}^*}(\mathbf{z}^{(k)})$ and $k \geq k_1$. It follows that there exists a sequence $\{m_k\}_{k \geq 1}$ such that $F(\mathbf{x}^{(m_k)}) > F_{\mathbf{m}^*}(\mathbf{z}^{(m_k)})$ with $m_k > k_1$ for all $k \geq 1$, and $\lim_{k \to \infty} m_k = \infty$. According to the PPGD algorithm and the Negative-Curvature-Exploitation algorithm described in Algorithm 1 and Algorithm 2, this is possible only if for all $k \geq 1$, there exists $i \in [d]$ such that $P(\mathbf{z}_i^{(m_k)}) \neq P(\mathbf{x}_i^{(m_k-1)}) = \mathbf{m}_i^*$, $\left| \mathbf{z}_i^{(m_k)} - q(\mathbf{w}_i^{(m_k-1)}) \right| < w_0 \left| \mathbf{z}_i^{(m_k)} - \mathbf{w}_i^{(m_k-1)} \right|$, $\mathbf{x}^{(m_k)} = \mathbf{x}^{(m_k-1)}$, and $f$ is continuous at $q(\mathbf{w}_i^{(m_k-1)})$.

We consider the case that $q(\mathbf{w}_i^{(m_k-1)}) = q_{\mathbf{m}_i^*}$ is the right endpoint of $\mathcal{R}_{\mathbf{m}_i^*}$. The case that $q(\mathbf{w}_i^{(m_k-1)}) = q_{\mathbf{m}_i^*-1}$ is the left endpoint of $\mathcal{R}_{\mathbf{m}_i^*}$ can be proved in a similar manner.

In this case, $P(\mathbf{z}_i^{(m_k)}) = \mathbf{m}_i^* + 1$. Let $\tilde{\mathbf{w}} \in \mathbb{R}^d$ with $\tilde{\mathbf{w}}_j = \mathbf{w}_j^{(m_k-1)}$ for all $j \neq i$, and $\tilde{\mathbf{w}}_i = q_{\mathbf{m}_i^*}$. By the definition of $\tilde{\mathbf{w}}$ and the same argument as (29) in the proof of Lemma B.2, we have

$$
\begin{aligned}
\mathbf{z}_i^{(m_k)} - \mathbf{w}_i^{(m_k-1)} &= -s\left([\nabla g(\mathbf{w}^{(m_k-1)})]_i + v^-\right) \\
&= s\left(\left[\nabla g(\mathbf{w}^{(m_k-1)}) - \nabla g(\tilde{\mathbf{w}})\right]_i + [\nabla g(\tilde{\mathbf{w}})]_i + v^-\right).
\end{aligned}
\tag{54}
$$

Because $\left\|\nabla g(\mathbf{w}^{(m_k-1)}) - \nabla g(\tilde{\mathbf{w}})\right\|_2 \leq sL_g(1-w_0)(G+F_0)$ and $s < \frac{\varepsilon_0}{L_g(1-w_0)(G+F_0)}$, we must have $[\nabla g(\tilde{\mathbf{w}})]_i + v^- \leq -\varepsilon_0$ due to Assumption 2 and the fact that $\mathbf{z}_i^{(m_k)} - \mathbf{w}_i^{(m_k-1)} > 0$.

With $k \to \infty$, we have $\frac{t_{m_k-1}}{t_{m_k}} \to 1$ and

$$
\begin{aligned}
\mathbf{u}^{(m_k)} &= \mathbf{x}^{(m_k)} + \frac{t_{m_k-1}}{t_{m_k}}(\mathbf{z}^{(m_k)} - \mathbf{x}^{(m_k)}) + \frac{t_{m_k-1}-1}{t_{m_k}}(\mathbf{x}^{(m_k)} - \mathbf{x}^{(m_k-1)}) \\
&= \mathbf{x}^{(m_k)} + \frac{t_{m_k-1}}{t_{m_k}}(\mathbf{z}^{(m_k)} - \mathbf{x}^{(m_k)}) \overset{k \to \infty}{\to} \mathbf{z}^{(m_k)},
\end{aligned}
\tag{55}
$$

and it follows that $\mathbf{w}^{(m_k)} = \mathbf{P}_{\mathbf{x}^{(m_k)}, R_0}(\mathbf{u}^{(m_k)}) \overset{k \to \infty}{\to} \mathbf{P}_{\mathbf{x}^{(m_k)}, R_0}(\mathbf{z}^{(m_k)})$, and we have $\left[\mathbf{P}_{\mathbf{x}^{(m_k)}, R_0}(\mathbf{z}^{(m_k)})\right]_i = \left[\mathbf{P}_{\mathbf{x}^{(m_k-1)}, R_0}(\mathbf{z}^{(m_k)})\right]_i = q_{\mathbf{m}_i^*}$. By the updating rule (9) in the PPGD algorithm and the first equality in (54), we have

$$
\left\|\mathbf{z}^{(m_k)} - \mathbf{w}^{(m_k-1)}\right\|_2 \leq s(G+\sqrt{d}F_0).
\tag{56}
$$

It follows from (56) that

$$
\begin{aligned}
\left\|\nabla g\left(\mathbf{P}_{\mathbf{x}^{(m_k)}, R_0}(\mathbf{z}^{(m_k)})\right) - \nabla g(\tilde{\mathbf{w}})\right\|_2 &\leq L_g \left\|\mathbf{P}_{\mathbf{x}^{(m_k)}, R_0}(\mathbf{z}^{(m_k)}) - \tilde{\mathbf{w}}\right\|_2 \\
&\overset{\text{①}}{\leq} L_g \left\|\mathbf{P}_{\mathbf{x}^{(m_k)}, R_0}(\mathbf{z}^{(m_k)}) - \mathbf{P}_{\mathbf{x}^{(m_k)}, R_0}(\mathbf{w}^{(m_k-1)})\right\|_2 \\
&\overset{\text{②}}{\leq} L_g \left\|\mathbf{z}^{(m_k)} - \mathbf{w}^{(m_k-1)}\right\|_2 \\
&\leq s L_g (G + \sqrt{d} F_0).
\end{aligned}
\tag{57}
$$

Here ① follows from $\left[\mathbf{P}_{\mathbf{x}^{(m_k)}, R_0}(\mathbf{z}^{(m_k)})\right]_i = \tilde{\mathbf{w}}_i = q_{\mathbf{m}_i^*}$, $\mathbf{w}^{(m_k-1)} = \mathbf{P}_{\mathbf{x}^{(m_k)}, R_0}(\mathbf{w}^{(m_k-1)})$, and $\mathbf{x}^{(m_k)} = \mathbf{x}^{(m_k-1)}$. ② follows from the contraction property of projection onto a closed convex set.

Combining (57) and the fact that $[\nabla g(\tilde{\mathbf{w}})]_i + v^- \leq -\varepsilon_0$, we have

$$
\begin{aligned}
\left[\nabla g\left(\mathbf{P}_{\mathbf{x}^{(m_k)}, R_0}(\mathbf{z}^{(m_k)})\right)\right]_i + v^- &= [\nabla g(\tilde{\mathbf{w}})]_i + v^- + \left(\left[\nabla g\left(\mathbf{P}_{\mathbf{x}^{(m_k)}, R_0}(\mathbf{z}^{(m_k)})\right)\right]_i - [\nabla g(\tilde{\mathbf{w}})]_i\right) \\
&\leq -\varepsilon_0 + s L_g (G + \sqrt{d} F_0) < 0
\end{aligned}
\tag{58}
$$

due to $s < \frac{\varepsilon_0}{L_g(G + \sqrt{d} F_0)}$.

Now noting that $\mathbf{w}^{(m_k)} = \mathbf{P}_{\mathbf{x}^{(m_k)}, R_0}(\mathbf{u}^{(m_k)}) \overset{k \to \infty}{\to} \mathbf{P}_{\mathbf{x}^{(m_k)}, R_0}(\mathbf{z}^{(m_k)})$, it follows from (58) and the smoothness of $\nabla g$ that there exists a large enough $k'$ such that when $k \geq k'$,

$$
\left[\nabla g(\mathbf{w}^{(m_k)})\right]_i + v^- < 0.
\tag{59}
$$

We now analyze the next iterate $\mathbf{z}_i^{(m_k+1)}$. By the updating rule $\mathbf{z}_i^{(m_k+1)} = \text{prox}_{s f_{P(\mathbf{x}_i^{(m_k)})}} \left(\left[\mathbf{w}^{(m_k)} - s\nabla g(\mathbf{w}^{(m_k)})\right]_i\right)$, we must have

$$
\mathbf{z}_i^{(m_k+1)} \in \mathcal{R}_{\mathbf{m}_i^*}^+, \quad \left[\mathbf{P}_{\mathbf{x}^{(m_k)}, R_0}(\mathbf{z}^{(m_k)})\right]_i = q_{\mathbf{m}_i^*} = q(\mathbf{w}_i^{(m_k)}).
\tag{60}
$$

In the iteration $m_k + 1$, we have

$$
d_{i,0} = \left|\mathbf{z}_i^{(m_k+1)} - \mathbf{w}_i^{(m_k)}\right|, d_{i,1} := \left|\mathbf{z}_i^{(m_k+1)} - q(\mathbf{w}_i^{(m_k)})\right|,
\tag{61}
$$

and $\mathbf{w}_i^{(m_k)} \overset{k \to \infty}{\to} q(\mathbf{w}_i^{(m_k)})$. Therefore, with sufficiently large $k'$, we have $d_{i,1} \geq w_0 d_{i,0}$ due to $d_{i,1} \overset{k \to \infty}{\to} d_{i,0}$. It follows from the Negative-Curvature-Exploitation algorithm described in Algorithm 2 that $P(\mathbf{x}_i^{(m_k+1)}) = \mathbf{m}_i^* + 1$, which contradict the fact that $P(\mathbf{x}^{(m_k)}) = \mathbf{m}^* \in \mathbb{N}^d$ for all $k \geq k_1$. This contradiction shows that there exists a finite $k_2 > k_1$ such that $F(\mathbf{x}^{(k)}) \leq F_{\mathbf{m}^*}(\mathbf{z}^{(k)})$ for all $k \geq k_2$. Setting $\bar{k} = k_2$ completes the proof.

$\square$

**Proof of Theorem 4.4.** According to Lemma B.5, there exists a finite $k_1 > 1$ such that $P(\mathbf{x}^{(k)}) = \mathbf{m}^* \in \mathbb{N}^d$ and $F(\mathbf{x}^{(k)}) \leq F_{\mathbf{m}^*}(\mathbf{z}^{(k)})$ for all $k \geq k_1$. Furthermore, the proof of Lemma 4.2 shows that the sequence $\{\mathbf{x}^{(k)}\}_{k \geq 1}$ generated by PPGD satisfies $\{\mathbf{x}^{(k)}\}_{k \geq 1} \subseteq \mathcal{L}$ which is a compact set, so there exists at least one limit point for $\{\mathbf{x}^{(k)}\}$, and $\Omega \neq \emptyset$.

Define $h_{\mathbf{m}}(\mathbf{x}) := \sum_{i=1}^d f_{\mathbf{m}_i}(\mathbf{x}_i)$ for $\mathbf{m} \in \mathbb{N}^d$ and $\mathbf{m}_i \in [M]$ for all $i \in [d]$, and $F_{\mathbf{m}} := g + h_{\mathbf{m}}$.

Note that for all $i \in [d]$, $f_{\mathbf{m}_i^*}$ is convex except for the third case in (5) or (6). In such a case, either event ①: $f_{\mathbf{m}_i^*}(x) = \lim_{y \to q_{\mathbf{m}_i^*}^+} f(y)$ for $x > q_{\mathbf{m}_i^*}$ and $f(q_{\mathbf{m}_i^*}) < \lim_{y \to q_{\mathbf{m}_i^*}^+} f(y)$, or event ②: $f_{\mathbf{m}_i^*}(x) = \lim_{y \to q_{\mathbf{m}_i^*-1}^-} f(y)$ for $x < q_{\mathbf{m}_i^*-1}$ and $f(q_{\mathbf{m}_i^*-1}) < \lim_{y \to q_{\mathbf{m}_i^*-1}^-} f(y)$. It follows by the proof of Lemma B.4 that all the sequences $\left\{\mathbf{x}_i^{(k)}\right\}_{k \geq k_1}$ and $\left\{\mathbf{z}_i^{(k)}\right\}_{k \geq k_1+1}$ satisfies $\left\{\mathbf{x}_i^{(k)}\right\}_{k \geq k_1} \subseteq (-\infty, q_{\mathbf{m}_i^*}]$ and $\left\{\mathbf{z}_i^{(k)}\right\}_{k \geq k_1+1} \subseteq (-\infty, q_{\mathbf{m}_i^*}]$ if event ① happens, and

$\left\{\mathbf{x}_i^{(k)}\right\}_{k \geq k_1} \subseteq [q_{\mathbf{m}_i^*-1}, +\infty)$ and $\left\{\mathbf{z}_i^{(k)}\right\}_{k \geq k_1+1} \subseteq [q_{\mathbf{m}_i^*-1}, +\infty)$ if event ② happens. For all $i \in [d]$, define $\mathcal{R}_i^*$ as the region over which $\tilde{f}_{\mathbf{m}_i^*}$ is convex. It is clear that $\mathcal{R}_i^* = \mathbb{R}$ if event ① and ② do not happen for $f_{\mathbf{m}_i^*}$. If only event ① happens for $f_{\mathbf{m}_i^*}$, then $\mathcal{R}_i^* = (-\infty, q_{\mathbf{m}_i^*}]$. If only event ② happens, $\mathcal{R}_i^* = [q_{\mathbf{m}_i^*-1}, +\infty)$. If both event ① and ② happen, then $\mathcal{R}_i^* = [q_{\mathbf{m}_i^*-1}, q_{\mathbf{m}_i^*}]$.

Let $\bar{\mathbf{x}} \in \mathbb{R}^d$ be an optimal solution to

$$\min_{\mathbf{x}_i \in \mathcal{R}_i^*, i \in [d]} F_{\mathbf{m}^*}(\mathbf{x}).$$

The existence of $\bar{\mathbf{x}}$ is proved as follows. First, it can be verified that the convex surrogate function $F_{\mathbf{m}^*}$ is continuous over the convex region $\mathcal{R}^*$, and $\mathcal{R}^*$ is a closed set in the usual Euclidean topology. By the coercivity assumption and the continuity of $F_{\mathbf{m}^*}$, the set $\mathcal{R}_0 := \left\{\mathbf{x} \mid F_{\mathbf{m}^*}(\mathbf{x}) \leq F_{\mathbf{m}^*}(\mathbf{x}^{(0)}), \mathbf{x} \in \mathcal{R}^*\right\}$ ($\mathbf{x}^{(0)}$ is the initialization point of PPGD) is bounded and closed, so the set $\mathcal{R}_0 \cap \mathcal{R}^*$ is bounded and closed thus a compact set. Therefore, the minimizer $\bar{\mathbf{x}}$ is by its definition a minimizer of $F_{\mathbf{m}^*}$ over $\mathcal{R}^*$, which is also a minimizer of a continuous function $F_{\mathbf{m}^*}$ over the compact set $\mathcal{R}_0 \cap \mathcal{R}^*$. The existence $\bar{\mathbf{x}}$ follows by the existence of a minimizer of a continuous function over a compact set.

**Roadmap of the proof.** We prove this theorem in three steps. In step 1, it is proved that there exists a finite $k_0 \geq k_1$ such that for all $k > k_0$,

$$F(\mathbf{x}^{(k)}) - F_{\mathbf{m}^*}(\bar{\mathbf{x}}) \leq \mathcal{O}(\frac{1}{k^2}).$$

Noting that $F(\mathbf{x}^{(k)}) = F_{\mathbf{m}^*}(\mathbf{x}^{(k)}) \geq F_{\mathbf{m}^*}(\bar{\mathbf{x}})$ by the optimality of $\bar{\mathbf{x}}$, the above inequality combined with the monotone nonincreasing of $\left\{F(\mathbf{x}^{(k)})\right\}$ indicate that $F(\mathbf{x}^{(k)}) \downarrow F_{\mathbf{m}^*}(\bar{\mathbf{x}})$.

In step 2, we will prove that $F(\mathbf{x}') = F_{\mathbf{m}^*}(\bar{\mathbf{x}})$ for any limit point $\mathbf{x}' \in \Omega$. According to the definition of $F_{\mathbf{m}^*}$ and the optimality of $\bar{\mathbf{x}}$, it follows that $F(\mathbf{x}')$ is a local minimum of $F$ and a global minimum of $F_{\mathbf{m}^*}$ over $\mathcal{R}^*$.

In step 3, we will prove that any limit point $\mathbf{x}' \in \Omega$ is a critical point of $F$ under a mild condition, following the argument in step 2.

**Step 1.** We now consider $k \geq k_1$ in the sequel. We have

$$f_{\mathbf{m}_i^*}(\mathbf{z}_i^{(k+1)}) \leq f_{\mathbf{m}_i^*}(v) + p(\mathbf{z}_i^{(k+1)} - v)$$

for all $i \in [d]$, $v \in \mathcal{R}_i$, and all $p \in \tilde{\partial} f(\mathbf{z}_i^{(k+1)})$. It follows that if $\mathbf{v} \in \mathbb{R}^d$ and $\mathbf{v}_i \in \mathcal{R}_i^*$ for all $i \in [d]$, then

$$h_{\mathbf{m}^*}(\mathbf{z}^{(k+1)}) \leq h_{\mathbf{m}^*}(\mathbf{v}) + \mathbf{p}(\mathbf{z}^{(k+1)} - \mathbf{v}) \tag{62}$$

for all $\mathbf{p} \in \tilde{\partial} h_{\mathbf{m}^*}$.

Because $\mathbf{z}_i^{(k+1)} = \text{prox}_{sf_{P(\mathbf{x}_i^{(k)})}}\left(\left[\mathbf{w}^{(k)} - s\nabla g(\mathbf{w}^{(k)})\right]_i\right)$ in (9) of Algorithm 1, it follows by the optimality of $\mathbf{z}_i^{(k+1)}$ that

$$\frac{1}{s}(\mathbf{w}^{(k)} - \mathbf{z}^{k+1}) - \nabla g(\mathbf{w}^k) \in \tilde{\partial} h_{\mathbf{m}^*}(\mathbf{z}^{(k+1)}). \tag{63}$$

It follows by (62) and (63) that

$$h_{\mathbf{m}^*}(\mathbf{z}^{(k+1)}) \leq h_{\mathbf{m}^*}(\mathbf{v}) + \left(\frac{1}{s}(\mathbf{w}^{(k)} - \mathbf{z}^{k+1}) - \nabla g(\mathbf{w}^k)\right)(\mathbf{z}^{(k+1)} - \mathbf{v}) \tag{64}$$

for any $\mathbf{v} \in \mathbb{R}^d$ such that $\mathbf{v}_i \in \mathcal{R}_i^*$ for all $i \in [d]$. For such $\mathbf{v}$, we have

$$
\begin{aligned}
F_{\mathbf{m}^*}(\mathbf{z}^{(k+1)}) &\leq g(\mathbf{v}) + \langle \nabla g(\mathbf{w}^{(k)}), \mathbf{z}^{(k+1)} - \mathbf{v} \rangle + \frac{L_g}{2} \|\mathbf{z}^{(k+1)} - \mathbf{w}^{(k)}\|_2^2 + h_{\mathbf{m}^*}(\mathbf{z}^{(k+1)}) \\
&\overset{①}{\leq} g(\mathbf{v}) + \langle \nabla g(\mathbf{w}^{(k)}), \mathbf{z}^{(k+1)} - \mathbf{v} \rangle + \frac{L_g}{2} \left\|\mathbf{z}^{(k+1)} - \mathbf{w}^{(k)}\right\|_2^2 + h_{\mathbf{m}^*}(\mathbf{v}) \\
&\quad + \langle \nabla g(\mathbf{w}^{(k)}) + \frac{1}{s}(\mathbf{z}^{(k+1)} - \mathbf{w}^{(k)}), \mathbf{v} - \mathbf{z}^{(k+1)} \rangle \\
&= F_{\mathbf{m}^*}(\mathbf{v}) + \frac{1}{s}\langle \mathbf{z}^{(k+1)} - \mathbf{w}^{(k)}, \mathbf{v} - \mathbf{z}^{(k+1)} \rangle + \frac{L_g}{2}\left\|\mathbf{z}^{(k+1)} - \mathbf{w}^{(k)}\right\|_2^2 \\
&\leq F_{\mathbf{m}^*}(\mathbf{v}) + \frac{1}{s}\langle \mathbf{z}^{(k+1)} - \mathbf{w}^{(k)}, \mathbf{v} - \mathbf{w}^{(k)} \rangle - \frac{1}{s}\|\mathbf{z}^{(k+1)} - \mathbf{w}^{(k)}\|_2^2 + \frac{L_g}{2}\left\|\mathbf{z}^{(k+1)} - \mathbf{w}^{(k)}\right\|_2^2 \\
&= F_{\mathbf{m}^*}(\mathbf{v}) + \frac{1}{s}\langle \mathbf{z}^{(k+1)} - \mathbf{w}^{(k)}, \mathbf{v} - \mathbf{w}^{(k)} \rangle - \left(\frac{1}{s} - \frac{L_g}{2}\right)\left\|\mathbf{z}^{(k+1)} - \mathbf{w}^{(k)}\right\|_2^2.
\end{aligned}
\tag{65}
$$

Here ① follows from (64).

Let $\mathbf{v} = \mathbf{x}^{(k)}$ and $\mathbf{v} = \bar{\mathbf{x}}$ in (65), we have

$$
F_{\mathbf{m}^*}(\mathbf{z}^{(k+1)}) \leq F_{\mathbf{m}^*}(\mathbf{x}^{(k)}) + \frac{1}{s}\langle \mathbf{z}^{(k+1)} - \mathbf{w}^{(k)}, \mathbf{x}^{(k)} - \mathbf{w}^{(k)} \rangle - \left(\frac{1}{s} - \frac{L_g}{2}\right)\left\|\mathbf{z}^{(k+1)} - \mathbf{w}^{(k)}\right\|_2^2,
\tag{66}
$$

and

$$
F_{\mathbf{m}^*}(\mathbf{z}^{(k+1)}) \leq F_{\mathbf{m}^*}(\bar{\mathbf{x}}) + \frac{1}{s}\langle \mathbf{z}^{(k+1)} - \mathbf{w}^{(k)}, \bar{\mathbf{x}} - \mathbf{w}^{(k)} \rangle - \left(\frac{1}{s} - \frac{L_g}{2}\right)\left\|\mathbf{z}^{(k+1)} - \mathbf{w}^{(k)}\right\|_2^2.
\tag{67}
$$

(66)$\times(t_k - 1)+$ (67), we have

$$
\begin{aligned}
&t_k F_{\mathbf{m}^*}(\mathbf{z}^{(k+1)}) - (t_k - 1)F_{\mathbf{m}^*}(\mathbf{x}^{(k)}) - F_{\mathbf{m}^*}(\bar{\mathbf{x}}) \\
&\leq \frac{1}{s}\langle \mathbf{z}^{(k+1)} - \mathbf{w}^{(k)}, (t_k - 1)(\mathbf{x}^{(k)} - \mathbf{w}^{(k)}) + \bar{\mathbf{x}} - \mathbf{w}^{(k)} \rangle - t_k\left(\frac{1}{s} - \frac{L_g}{2}\right)\left\|\mathbf{z}^{(k+1)} - \mathbf{w}^{(k)}\right\|_2^2.
\end{aligned}
\tag{68}
$$

It follows that

$$
\begin{aligned}
&t_k\big(F_{\mathbf{m}^*}(\mathbf{z}^{(k+1)}) - F_{\mathbf{m}^*}(\bar{\mathbf{x}})\big) - (t_k - 1)\big(F_{\mathbf{m}^*}(\mathbf{x}^{(k)}) - F_{\mathbf{m}^*}(\bar{\mathbf{x}})\big) \\
&\leq \frac{1}{s}\langle \mathbf{z}^{(k+1)} - \mathbf{w}^{(k)}, (t_k - 1)(\mathbf{x}^{(k)} - \mathbf{w}^{(k)}) + \bar{\mathbf{x}} - \mathbf{w}^{(k)} \rangle - t_k\left(\frac{1}{s} - \frac{L_g}{2}\right)\left\|\mathbf{z}^{(k+1)} - \mathbf{w}^{(k)}\right\|_2^2.
\end{aligned}
\tag{69}
$$

Multiplying both sides of (69) by $t_k$, since $t_k^2 - t_k = t_{k-1}^2$, we have

$$
\begin{aligned}
&t_k^2\left(F_{\mathbf{m}^*}(\mathbf{z}^{(k+1)}) - F_{\mathbf{m}^*}(\bar{\mathbf{x}})\right) - t_{k-1}^2\left(F_{\mathbf{m}^*}(\mathbf{x}^{(k)}) - F_{\mathbf{m}^*}(\bar{\mathbf{x}})\right) \\
&\leq \frac{1}{s}\langle t_k(\mathbf{z}^{(k+1)} - \mathbf{w}^{(k)}), (t_k - 1)(\mathbf{x}^{(k)} - \mathbf{w}^{(k)}) + \bar{\mathbf{x}} - \mathbf{w}^{(k)} \rangle - (\frac{1}{s} - \frac{L_g}{2})\|t_k(\mathbf{z}^{(k+1)} - \mathbf{w}^{(k)})\|_2^2 \\
&\leq \frac{1}{s}\langle t_k(\mathbf{z}^{(k+1)} - \mathbf{w}^{(k)}), (t_k - 1)(\mathbf{x}^{(k)} - \mathbf{w}^{(k)}) + \bar{\mathbf{x}} - \mathbf{w}^{(k)} \rangle - \frac{1}{2s}\left\|t_k(\mathbf{z}^{(k+1)} - \mathbf{w}^{(k)})\right\|_2^2 \\
&= \frac{1}{2s}\left(\left\|(t_k - 1)\mathbf{x}^{(k)} - t_k\mathbf{w}^{(k)} + \bar{\mathbf{x}}\right\|_2^2 - \left\|(t_k - 1)\mathbf{x}^{(k)} - t_k\mathbf{z}^{(k+1)} + \bar{\mathbf{x}}\right\|_2^2\right).
\end{aligned}
\tag{70}
$$

Since $t_k \geq \frac{k+1}{2}$ for $k \geq 1$, we have $t_k \overset{k\to\infty}{\to} \infty$ and $\lim_{k\to\infty}\left(1 - \frac{1}{t_k}\right)\mathbf{x}^{(k)} + \frac{1}{t_k}\bar{\mathbf{x}} = \mathbf{x}^{(k)}$. It follows that there exists a finite $k_2$ such that

$$
\left[\left(1 - \frac{1}{t_k}\right)\mathbf{x}^{(k)} + \frac{1}{t_k}\bar{\mathbf{x}}\right]_i \in \overline{\mathcal{R}_{P(\mathbf{x}_i^{(k)})}} \cap B(\mathbf{x}_i^{(k)}, R_0)
\tag{71}
$$

for all $k \geq k_2$ and all $i \in [d]$. It follows from (71) that

$$
\left(1 - \frac{1}{t_k}\right)\mathbf{x}^{(k)} + \frac{1}{t_k}\bar{\mathbf{x}} = \mathbf{P}_{\mathbf{x}^{(k)}, R_0}\left(\left(1 - \frac{1}{t_k}\right)\mathbf{x}^{(k)} + \frac{1}{t_k}\bar{\mathbf{x}}\right).
\tag{72}
$$

Now let $k \geq k_0 := \max\{k_1, k_2\}$, we have

$$
\begin{aligned}
\left\| (t_k - 1)\mathbf{x}^{(k)} - t_k \mathbf{w}^{(k)} + \bar{\mathbf{x}} \right\|_2 &= t_k \left\| \left(1 - \frac{1}{t_k}\right)\mathbf{x}^{(k)} + \frac{1}{t_k}\bar{\mathbf{x}} - \mathbf{w}^{(k)} \right\|_2 \\
&\overset{\textcircled{1}}{=} t_k \left\| \mathbf{P}_{\mathbf{x}^{(k)}, R_0}\left( \left(1 - \frac{1}{t_k}\right)\mathbf{x}^{(k)} + \frac{1}{t_k}\bar{\mathbf{x}} \right) - \mathbf{P}_{\mathbf{x}^{(k)}, R_0}(\mathbf{u}^{(k)}) \right\|_2 \\
&\overset{\textcircled{2}}{\leq} t_k \left\| \left(1 - \frac{1}{t_k}\right)\mathbf{x}^{(k)} + \frac{1}{t_k}\bar{\mathbf{x}} - \mathbf{u}^{(k)} \right\|_2 \\
&\leq \left\| (t_k - 1)\mathbf{x}^{(k)} - t_k \mathbf{u}^{(k)} + \bar{\mathbf{x}} \right\|_2,
\end{aligned}
\tag{73}
$$

where $\textcircled{1}$ follows from (72), $\textcircled{2}$ follows from the contraction property of projection onto a closed convex set.

It follows by (70) and (73), that

$$
\begin{aligned}
&t_k^2 \left( F_{\mathbf{m}^*}(\mathbf{z}^{(k+1)}) - F_{\mathbf{m}^*}(\bar{\mathbf{x}}) \right) - t_{k-1}^2 \left( F_{\mathbf{m}^*}(\mathbf{x}^{(k)}) - F_{\mathbf{m}^*}(\bar{\mathbf{x}}) \right) \\
&\leq \frac{1}{2s} \left( \left\| (t_k - 1)\mathbf{x}^{(k)} - t_k \mathbf{u}^{(k)} + \bar{\mathbf{x}} \right\|_2^2 - \left\| (t_k - 1)\mathbf{x}^{(k)} - t_k \mathbf{z}^{(k+1)} + \bar{\mathbf{x}} \right\|_2^2 \right).
\end{aligned}
\tag{74}
$$

Define $\mathbf{Q}^{(k+1)} = (t_k - 1)\mathbf{x}^{(k)} - t_k \mathbf{z}^{(k+1)} + \bar{\mathbf{x}}$, then $\mathbf{Q}^{(k)} = (t_{k-1} - 1)\mathbf{x}^{(k-1)} - t_{k-1}\mathbf{z}^{(k)} + \bar{\mathbf{x}}$. It can be verified that $\mathbf{Q}^{(k)} = (t_k - 1)\mathbf{x}^{(k)} - t_k \mathbf{u}^{(k)} + \bar{\mathbf{x}}$. Therefore,

$$
t_k^2 \left( F_{\mathbf{m}^*}(\mathbf{z}^{(k+1)}) - F_{\mathbf{m}^*}(\bar{\mathbf{x}}) \right) - t_{k-1}^2 \left( F_{\mathbf{m}^*}(\mathbf{x}^{(k)}) - F_{\mathbf{m}^*}(\bar{\mathbf{x}}) \right) \leq \frac{1}{2s} \left( \left\| \mathbf{Q}^{(k)} \right\|_2^2 - \left\| \mathbf{Q}^{(k+1)} \right\|_2^2 \right).
$$

Noting that $F(\mathbf{x}^{(k+1)}) \leq F_{\mathbf{m}^*}(\mathbf{z}^{(k+1)})$ for $k \geq k_0$, it follows from the above inequality that

$$
t_k^2 \left( F_{\mathbf{m}^*}(\mathbf{x}^{(k+1)}) - F_{\mathbf{m}^*}(\bar{\mathbf{x}}) \right) - t_{k-1}^2 \left( F_{\mathbf{m}^*}(\mathbf{x}^{(k)}) - F_{\mathbf{m}^*}(\bar{\mathbf{x}}) \right) \leq \frac{1}{2s} \left( \left\| \mathbf{Q}^{(k)} \right\|_2^2 - \left\| \mathbf{Q}^{(k+1)} \right\|_2^2 \right).
\tag{75}
$$

Summing (75) over $k = k_0, \ldots, m$ for $m \geq k_0$, we have

$$
\begin{aligned}
&t_m^2 \left( F_{\mathbf{m}^*}(\mathbf{x}^{(m+1)}) - F_{\mathbf{m}^*}(\bar{\mathbf{x}}) \right) - t_{k_0-1}^2 \left( F_{\mathbf{m}^*}(\mathbf{x}^{(k_0)}) - F_{\mathbf{m}^*}(\bar{\mathbf{x}}) \right) \\
&\leq \frac{1}{2s} \left( \|\mathbf{Q}^{(k_0)}\|_2^2 - \|\mathbf{Q}^{(m+1)}\|_2^2 \right) \leq \frac{1}{2s} \left\| \mathbf{Q}^{(k_0)} \right\|_2^2 = \frac{1}{2s} \left\| (t_{k_0-1} - 1)\mathbf{x}^{(k_0-1)} - t_{k_0-1}\mathbf{z}^{(k_0)} + \bar{\mathbf{x}} \right\|_2^2.
\end{aligned}
\tag{76}
$$

Since $t_k \geq \frac{k+1}{2}$ for $k \geq 1$, it follows from (76) that

$$
\begin{aligned}
F_{\mathbf{m}^*}(\mathbf{x}^{(m+1)}) - F_{\mathbf{m}^*}(\bar{\mathbf{x}}) &\leq \frac{4}{(m+1)^2} \left( \frac{1}{2s} \left\| (t_{k_0-1} - 1)\mathbf{x}^{(k_0-1)} - t_{k_0-1}\mathbf{z}^{(k_0)} + \bar{\mathbf{x}} \right\|_2^2 \right. \\
&\qquad\qquad \left. + t_{k_0-1}^2 \left( F_{\mathbf{m}^*}(\mathbf{x}^{(k_0)}) - F_{\mathbf{m}^*}(\bar{\mathbf{x}}) \right) \right) \\
&\triangleq \frac{4}{(m+1)^2} U^{(k_0)}.
\end{aligned}
\tag{77}
$$

Noting that $F(\mathbf{x}^{(m+1)}) = F_{\mathbf{m}^*}(\mathbf{x}^{(m+1)})$, we have $F(\mathbf{x}^{(m+1)}) - F_{\mathbf{m}^*}(\bar{\mathbf{x}}) \leq \frac{4}{(m+1)^2} U^{(k_0)}$. Replacing $m + 1$ with $k$, we have

$$
F(\mathbf{x}^{(k)}) - F_{\mathbf{m}^*}(\bar{\mathbf{x}}) \leq \frac{4}{k^2} U^{(k_0)}
\tag{78}
$$

for all $k > k_0$. Because $F(\mathbf{x}^{(k)}) = F_{\mathbf{m}^*}(\mathbf{x}^{(k)}) \geq F_{\mathbf{m}^*}(\bar{\mathbf{x}})$ due to the optimality of $\bar{\mathbf{x}}$, we have $F(\mathbf{x}^{(k)}) \downarrow F_{\mathbf{m}^*}(\bar{\mathbf{x}})$ as $k \to \infty$ based on (78).

**Step 2.** We now prove that any limit point $\mathbf{x}' \in \Omega$ achieves a local minimum of $F$. Let $\mathbf{x}' \in \Omega$ be an arbitrary limit point of $\{\mathbf{x}^{(k)}\}_{k \geq 1}$. By Lemma 4.2, we have $F(\mathbf{x}^{(k)}) \downarrow F(\mathbf{x}')$ as $k \to \infty$. To see this, we first note that $F_{\mathbf{m}^*}$ is continuous over the set $\mathcal{R}^*$ by the definition of $\mathcal{R}^*$ in the beginning of this proof. It follows by the beginning of this proof that $\left\{\mathbf{x}_i^{(k)}\right\}_{k \geq k_0} \subseteq \mathcal{R}_i^*$ for all $k \geq k_0$ and all $i \in [d]$. In addition, $\mathbf{x}_i' \in \mathcal{R}_{\mathbf{m}_i^*} \subseteq \mathcal{R}_i^*$ for all $i \in [d]$. Therefore, $\{\mathbf{x}^{(k)}\}_{k \geq k_0}$ and $\mathbf{x}'$ belong to $\mathcal{R}^*$ on which $F_{\mathbf{m}^*}$ is continuous, so $F(\mathbf{x}^{(k)}) = F_{\mathbf{m}^*}(\mathbf{x}^{(k)}) \downarrow F_{\mathbf{m}^*}(\mathbf{x}') = F(\mathbf{x}')$.

We also have $F(\mathbf{x}^{(k)}) \downarrow F_{\mathbf{m}^*}(\bar{\mathbf{x}})$ as $k \to \infty$ due to step 1. As a result,

$$F(\mathbf{x}') = F_{\mathbf{m}^*}(\bar{\mathbf{x}}) \tag{79}$$

for any $\mathbf{x}' \in \Omega$. That is, $F$ has constant value on $\Omega$.

It is noted that $F_{\mathbf{m}^*} = F$ on the set $\{\mathbf{x} \in \mathbb{R}^d \,|\, P(\mathbf{x}) = \mathbf{m}^*\} \subseteq \mathcal{R}^*$, so the optimality of $\bar{\mathbf{x}}$ indicates that

$$F(\mathbf{x}') = F_{\mathbf{m}^*}(\bar{\mathbf{x}}) \leq \inf_{\{\mathbf{x} \in \mathbb{R}^d \,|\, P(\mathbf{x}) = \mathbf{m}^*\}} F(\mathbf{x}). \tag{80}$$

On the other hand, since $P(\bar{\mathbf{x}}) = \mathbf{m}^*$, we have $F(\mathbf{x}') \geq \inf_{\{\mathbf{x} \in \mathbb{R}^d \,|\, P(\mathbf{x}) = \mathbf{m}^*\}} F(\mathbf{x})$. Combining this inequality and (80), we have $F(\mathbf{x}') = \inf_{\{\mathbf{x} \in \mathbb{R}^d \,|\, P(\mathbf{x}) = \mathbf{m}^*\}} F(\mathbf{x})$.

**Step 3.** We now prove that any limit point $\mathbf{x}' \in \Omega$ is a critical point of $F$ under a mild condition that $f_{\mathbf{m}_i^*}$ does not take the third case in (5) or (6) for all $i \in [d]$. As explained in the beginning of this proof, under this condition, $\mathcal{R}_i^* = \mathbb{R}$ for all $i \in [d]$. Because $F(\mathbf{x}') = F_{\mathbf{m}^*}(\bar{\mathbf{x}})$ for any $\mathbf{x}' \in \Omega$, $\mathbf{x}'$ is an optimal solution to

$$\min_{\mathbf{x} \in \mathbb{R}^d} F_{\mathbf{m}^*}(\mathbf{x}),$$

and $F_{\mathbf{m}^*}$ is convex over $\mathbb{R}^d$. The optimality of $\mathbf{x}'$ for this convex programming problem indicates that $\mathbf{0} \in \tilde{\partial} F_{\mathbf{m}^*}(\mathbf{x}')$. Because $F_{\mathbf{m}^*} = F$ on the set $\{\mathbf{x} \in \mathbb{R}^d \,|\, P(\mathbf{x}) = \mathbf{m}^*\}$ and $\mathbf{x}' \in \{\mathbf{x} \in \mathbb{R}^d \,|\, P(\mathbf{x}) = \mathbf{m}^*\}$, we have $\mathbf{0} \in \partial F(\mathbf{x}')$. This can be verified using the definition of limiting subdifferential by considering a constant sequence $\mathbf{x}^k = \mathbf{x}'$ for all $k \geq 1$. We have with $0 \in \tilde{\partial} F(\mathbf{x}^k)$ because $\mathbf{0} \in \tilde{\partial} F_{\mathbf{m}^*}(\mathbf{x}')$ and $P(\mathbf{x}') = \mathbf{m}^*$.

$\square$

