# OpenReview forum: "Projective Proximal Gradient Descent for Nonconvex Nonsmooth Optimization: Fast Convergence Without Kurdyka-Lojasiewicz (KL) Property"
_ICLR.cc/2023/Conference — ICLR 2023 poster_

### Official Review · Reviewer_tNXt · 2022-10-19

**Confidence:** 4
**Correctness:** 3
**Technical Novelty And Significance:** 4
**Empirical Novelty And Significance:** Not applicable
**Recommendation:** 8

**Clarity, Quality, Novelty And Reproducibility:**

***Clarity***
The paper is overall speaking clear. Below are some issues.
- What is the point of showing Algorithm 1 in the paper?
- Should there be a condition $w_0 < 1 / ( G + F )$?
- An explanation or definition of the parameter $w_0$ in Algorithm 3 seems to be missing.
- p. 1: Let $h_j = f$ should not be "without loss of generality" but for simplicity.
- p. 2: The symbol for the bound on the Frechet subdifferential conflicts with the symbol for the objective function. Both are $F$ in the paper.

***Quality***
The paper is overall speaking readable. Below are some suggestions.
- Assumption 1 and the definition of $J$ are quite technical and can be moved to later sections.
- There is no need to provide the complete definition of the KL property.
- Algorithm 1 can be removed or moved to the appendix.

The paper seems to be written in haste. There are some typos, such as:
- Abstract: nd -> and
- Assumption 1(c): has close form solution -> has a closed-form solution
- Assumption 1(d): Some words are missing between "at" and "for."
- p. 3: The norm should be removed in (4).
- p. 3: In the first sentence of Section 1.2, there are some words missing between "are" and "The."
- p. 7: "*the an* important Theorem"

***Novelty***
The algorithm and its analysis are novel.

***Reproducibility***
The codes for the numerical results are not provided. But this is a paper of theorems, so I don't think this is a big issue.

**Strength And Weaknesses:**

***Strength***
1. The problem formulation is relevant to high-dimensional statisticians.
2. The authors took a respectful effort handling possibly discontinuous objective functions.
3. The proposed method is the fastest in the numerical experiments.

***Weaknesses***
1. The main issue lies in the definition of "convergence." The convergence guarantee (Theorem 3.5) only guarantees the difference between the function values at the iterates and *any limit point of the iterates* vanish at a $O ( 1 / k ^ 2 )$ rate. Nevertheless, a limit point does not obviously possess any optimality property. The meaning of the convergence guarantee is hence unclear.
2. Unlike the iteration complexity analysis for Nesterov's accelerated gradient descent, this paper needs an additional assumption that the convex smooth function is also Lipschitz. The claimed optimality of the $O ( 1 / k^2 )$ rate is hence unclear.
3. The existence of the operator $P$, which returns the index of the convex piece of an inquiry point, does not seem to be efficiently implementable in general. That the operator is efficiently implementable should be put as an assumption.

**Summary Of The Paper:**

Consider the problem of minimizing the sum of a convex smooth and Lipschitz function and a separable piecewise-convex Lipschitz regularizer. This paper proposes an algorithm, based on Nesterov's accelerated gradient method and a "negative curvature exploitation" procedure, that converges at a $O ( 1 / k^2 )$ rate. The key idea is to design the negative curvature exploitation procedure, such that every time the iterate leaves a convex piece, the objective function value is decreased by at least a constant. Then, asymptotically all iterates lie in the same convex piece and the known convergence rate of Nesterov's accelerated gradient method applies.

**Summary Of The Review:**

The problem formulation is relevant and the algorithm and its analysis are novel. The presentation has some space for improvement but is overall speaking acceptable. The main issue lies in the convergence guarantee (Theorem 3.5), which only guarantees a $O ( 1 / k ^ 2 )$ asymptotic convergence rate to the function value at a limit point, but the limit point does not have any obvious optimality guarantee. As the meaning of the main theoretical result is unclear, I cannot suggest acceptance of this paper in its current form.

***After discussion with authors***
In the latest revision (https://github.com/iclrpaper5324/PPGD/blob/main/ppgd.pdf), the authors have fixed the three weaknesses I pointed out, so I change my recommendation to "accept." *However, I am not sure if making decisions based on a version of the submission not on OpenReview is acceptable.*

The details are below.
- In Step 2 of the proof of Theorem 4.4, it has been proved that any limit point is a local minimum of $F$ on the final convex pieces.
- In Assumption 1, the Lipschitz assumption has been replaced by a coercivity assumption of $F$.
- Computation of the projection operator has been addressed in the author response ((3) Efficient Computation of the Operator $P$).

One comment on the proof of Theorem 4.4: That the objective function and the constraint set are both convex does not necessarily imply existence of the minimizer $\bar{x}$. I think a coercivity argument is needed.

---

> ### Author Response · Authors · 2022-11-20
> **Response to Reviewer tNXt Part 1**
>
> We appreciate the review and the suggestions in this review. The raised issues are addressed below.
>
> (1) **Completed Convergence Results of PPGD**
>
> The major theorem, Theorem 4.4 in the revised paper, presents the completed convergence results of PPGD with the optimality of the limit points the iterates of PPGD converge to. Theorem 4.4 is copied below for your convenience which is followed by its explanation.
>
> **Theorem 4.4**
>
> Suppose the step size $s < \min\set{s_1,\frac{\epsilon_0}{L_g (G+\sqrt{d}F_0)},\frac{1}{L_g}}$ with $s_1$ defined by eq. (16),  Assumption 1 and Assumption 2 hold. Then there exists a finite $k_0 \ge 1$ such that the following statements hold. (1) $P(\mathbf x^{(k)}) = \mathbf m^*$ for some $\mathbf m^* \in \mathbb N^d$ for all $k > k_0$. (2) Let $\Omega \coloneqq
> \left( \mathbf x \mid \mathbf x \textup{ is a limit point of }  (\mathbf x^{(k)}), P(\mathbf x) = \mathbf m^* \right)$ be the set of all limit points of the sequence $\left(\mathbf x^{(k)}\right)$ generated by Algorithm 1lying on the convex pieces indexed by $\mathbf m^*$. Then for any $\mathbf x^* \in \Omega$, $F(\mathbf x^*) = \inf\limits_{\left(\mathbf x \in \mathbb R^d \mid P(\mathbf x) = \mathbf m^* \right)} F(\mathbf x)$, and
>
> $F(\mathbf x^{(k)})-F(\mathbf x^*) \le \frac{4}{k^2} U^{(k_0)} \quad (18)$
>
> for all $k > k_0$, where $U^{(k_0)} \coloneqq \left( \frac{1}{2s} \lvert\lvert (t_{k_0-1}-1) \mathbf x^{(k_0-1)} - t_{k_0-1} \mathbf z^{(k_0)} + \mathbf x^* \rvert\rvert_2^2 + t_{k_0-1}^2 ( F(\mathbf x^{(k_0)})-F(\mathbf x^*) ) \right)$. Moreover, if $f_{\mathbf m_i^*}$ does not take the third case in (5) or (6) for all $i \in [d]$, then $\mathbf x^*$ is a critical point of $F$, that is, $0 \in \partial F(\mathbf x^*)$.
>
> **Explanation of Theorem 4.4** Theorem 4.4 states that after a finite number ($k_0$) of iterations, all iterates $\left(\mathbf x^{(k)}\right)$ with $k > k_0$ are on the same convex pieces indexed by $\mathbf m^*$, and any limit point $\mathbf x^*$ of the sequence $ (\mathbf x^{(k)})$ on such convex pieces has the following nice properties. (1) $\mathbf x^*$ is a local minimum of the objective function $F$ on the convex pieces indexed by $\mathbf m^*$, that is, $F(\mathbf x^*) = \inf\limits_{\left(\mathbf x \in \mathbb R^d \mid P(\mathbf x) = \mathbf m^* \right)} F(\mathbf x)$. (2) $F(\mathbf x^{(k)})$ converges to $ F(\mathbf x^*) $ at a rate of $\mathcal O(1/k^2)$, which locally matches the Nesterov’s optimal convergence rate of first-order methods on smooth and convex objective function with Lipschitz continuous gradient. (3) $\mathbf x^*$ is a critical point of $F$ if $f_{\mathbf m_i^*}$ does not take the third case in (5) or (6) for all $i \in [d]$, that is, the surrogate function $f_{\mathbf m_i^*}$ does not have “jump-up” discontinuity, or formally, both $f(q_{\mathbf m_i^*}) < \lim_{y \to q_{\mathbf m_i^*}^+} f(y)$ and $f(q_{\mathbf m_i^*-1}) < \lim_{y \to q_{\mathbf m_i^*-1}^-} f(y)$ do not happenfor all $i \in [d]$. This scenario happens for all continuous $f$, for example, capped-$\ell^1$ or leaky capped-$\ell^1$ penalty functions.
>
> (2) **The Claimed Optimality of $\mathcal O(\frac{1}{k^2})$**
>
> We have removed the assumption about bounded gradient, that is, $\sup_{\mathbf x \in \mathbb R^d} \vert\vert g(\mathbf x) \vert\vert \le G$, in Assumption 1 of the revised paper. Instead, the commonly adopted coercive assumption is adopted. All the theoretical and empirical results of this paper still hold. Please refer to **Relaxed Assumptions** in Part 1 of our response to Reviewer Dzxh for details. As a result, we do not require the convex smooth function $g$ is Lipschitz in the revised paper. We have also changed the statement about the optimal convergence rate achieved by PPGD and emphasized that PPGD **locally** matches the Nesterov’s optimal convergence rate, in the sense that the convergence rate of $\mathcal O(\frac{1}{k^2})$, that is, $F(\mathbf x^{(k)})-F(\mathbf x^*) \le \frac{4}{k^2} U^{(k_0)} $, is achieved when $k \ge k_0$ for some finite $k_0$. We respectfully point out that the local rate $\mathcal O(\frac{1}{k^2})$ is locally optimal, because all the iterates $\left( \mathbf x^{(k)}\right)$ for $k > k_0$ are on the same convex pieces indexed by $ \mathbf m^*$, and restricted on such convex pieces the objective $F$ is smooth and convex where $\mathcal O(\frac{1}{k^2})$ is known to be an optimal rate.

---

> > ### Author Response · Authors · 2022-11-20
> > **Response to Reviewer tNXt Part 2**
> >
> > (3) **Efficient Computation of the Operator $P(\cdot)$**
> >
> > We respectfully point out that it is not necessary to assume $P$ is efficiently implementable, because $P$ can be computed efficiently as follows. The endpoints of the convex pieces, $\left(q_m\right)$ for $1 \le m \le M-1$ sorted in ascending order, are given as the input to the PPGD described in Algorithm 1 of the revised paper. The projection $P(x)$, which is the index of the convex piece that $x$ belongs to, can be computed efficiently by comparing $x$ to the endpoints. In particular, let $m$ be the smallest index such that $x \le q_m$. If $x < q_m$, then $P(x)=m$. If $x = q_m$, then $P(x) = m+1$ if $f(q_m) < \lim_{y \to q_m-} f(y)$, and $P(x) = m$ otherwise. $f(q_m) < \lim_{y \to q_m-} f(y)$ can be efficiently implemented by comparing $ f(q_m)$ to $f(y)$ with $y < q_m$ but sufficiently close to $q_m$.
> >
> > We have addressed the remaining issues and revised the paper according to your suggestions. The original Algorithm 1 and the definition of the KL property have been moved to the supplementary. $w_0$ is specified as an input to the PPGD described in Algorithm 1 of the revised paper. The bound for the Frechet subdifferential has been changed to $F_0$. Assumption 1 and the definition of $J$ have been moved to a dedicated subsubsection 1.1 of the revised paper. we have also fixed all the grammatical issues and typos in the revised paper.
> >
> > **Improved Presentation**
> > We have significantly improved the presentation of this paper with more introduction to the background and intuition and motivation of PPGD. Please kindly refer to “(2) **The Intuition and Motivation to Construct Surrogate Functions, and Improved Presentation**” in Part 1 of our response to Reviewer ofLD for more details.

---

> > ### Comment · Reviewer_tNXt · 2022-12-08
> > **Re: Response to Reviewer tNXt Part 1**
> >
> > Dear authors,
> >
> > Sorry for the late response.
> >
> > I have seen that you removed the Lipschitz assumption on $g$ and introduce a new assumption on the coercivity on the whole objective function.
> >
> > As for the main theorem, could you point out how you modified the proof to obtain an improved convergence guarantee? This would save a lot of my time.

---

> > > ### Author Response · Authors · 2022-12-08
> > > **Roadmap of Proof of Main Theorem (Theorem 4.4) with Indicated Modifications, and the Locally Optimal Convergence Rate with the Coercivity Assumption Part 1**
> > >
> > > Dear Reviewer tNXt,
> > >
> > > Thank you for your response! We present the intuition behind the proof of the main theorem (Theorem 4.4) and its detailed technical roadmap where the modifications are indicated. We also explain that the convergence rate $\mathcal O(\frac{1}{k^2})$  of PPGD is still locally optimal under the coercivity assumption.
> > >
> > > It is noted that the majority of the proof was already in the originally submitted paper, and the limit point in the original proof already enjoyed the optimality properties. We made minor changes in the proof of Theorem 4.4 of the revised paper to fully reveal the optimality of the limit points. We first introduce the definitions in the original paper and the revised paper, and then present the intuition and the roadmap of the proof. You are also cordially invited to view the slightly edited proof of Theorem 4.4 through the anonymous link https://github.com/iclrpaper5324/PPGD/blob/main/ppgd.pdf, where the details of the roadmap below are added. Such minor edits will be reflected in the final version of the paper.
> > >
> > > **Definition in the Original Paper**
> > >
> > > We define $F_{\mathbf m}(\mathbf x) \coloneqq g(\mathbf x) + \sum_{i=1}^d f_{\mathbf m_i}(\mathbf x_i)$ for any $\mathbf m \in \mathbb N^d$ and all $\mathbf x \in \mathbb R^d$, which is the surrogate objective function on convex pieces indexed by $\mathbf m$.   Let $\mathcal R_i^*$  be the region on which $f_{\mathbf m_i^*}$ is convex, then $F_{\mathbf m^*}$ is convex over the set $\mathcal R^* \coloneqq \left(\mathbf x \in \mathbb R^d \mid \mathbf x_i \in \mathcal R_i^*, \forall i \in [d] \right)$. It is proved in the original paper (in the beginning of proof of Theorem 3.4 of the original paper, with $\mathcal R_i$  replaced by $\mathcal R_i^*$) that $\left( \mathbf x^{(k)}\right) \subseteq \mathcal R^*$ for all $k \ge k_0$.
> > >
> > > **Definition in the Revised Paper**
> > >
> > > Let $\bar{\mathbf x}$ be an optimal solution to $\min\limits_{\mathbf x \in \mathbb R^d, \mathbf x_i \in \mathcal R_i^*} F_{\mathbf m^*}(\mathbf x)$. We also define
> > >
> > > $\Omega \coloneqq \left( \mathbf x \mid \mathbf x \textup{ is a limit point of } \left(\mathbf x^{(k)}\right)_{k \ge 1}, P(\mathbf x) = \mathbf m^* \right)$
> > >
> > > be the set of all limit points of the sequence $\left(\mathbf x^{(k)}\right)_{k \ge 1}$ generated by the PPGD algorithm lying on the convex pieces indexed by $\mathbf m^*$.
> > >
> > >
> > > **Intuition**
> > >
> > > The key idea of proof is based on an important result, which was proved in the original paper, that there exists a finite $k_0 \ge 1$ such that all the iterates $\left( \mathbf x^{(k)}\right)$ for $k > k_0$ are on the same convex pieces indexed by $\mathbf m^*$. Therefore, the optimization of the original objective function $F$ by PPGD is the same as optimization of the convex surrogate objective function $F_{\mathbf m^*}$. Using the same arguments in the original paper (contraction property of our novel projection operator $\mathbf P$ defined in eq. (11) and the standard arguments in the proof of convergence rate for convex optimization problems), it is proved that $F(\mathbf x^{(k)}) \downarrow F_{\mathbf m^*}(\bar{\mathbf x})$. $F_{\mathbf m^*}(\bar{\mathbf x})$ is, by definition, a local minimum of $F$ and a global minimum of $F_{\mathbf m^*}$. As a result, all the nice optimality properties of the limit points in $\Omega$ hold (detailed in step 2 of the roadmap). The derivations are detailed in the roadmap below.
> > >
> > > **Roadmap of Proof of Theorem 4.4** (modifications in the revised paper are indicated)
> > >
> > > **Step 1. (in the original paper)**  We prove $F(\mathbf x^{(k)}) \downarrow F_{\mathbf m^*}(\bar{\mathbf x})$  using the same arguments in the proof of Theorem 3.4 in the original paper (by replacing $\mathbf x^*$ in the original proof with $\bar{\mathbf x}$), and the locally optimal convergence rate is $\mathcal O(\frac{1}{k^2})$ for $k > k_0$.  It is noted that the contraction property of the projection operator $\mathbf P$ leads to eq. (69) in the revised paper in the same manner as eq. (69) in the original paper.
> > >
> > > **Step 2. (modification in the revised paper), proving any limit point $\mathbf x' \in \Omega$ is a local mininum of $F$**. By Lemma 4.2 in the revised paper, $\left(F(\mathbf x^{(k)})\right)$  is a nonincreasing sequence. We have the following results. (1) $F_{\mathbf m^*}$ is continuous over the set $\mathcal R^*$. (2) $\left( \mathbf x_i^{(k)}\right) \subseteq \mathcal R_i$ holds for all $k \ge k_0$ and $i \in [d]$. (3) Because $P(\mathbf x'_i) = \mathbf m_i^*$, we have $\mathbf x'_i \in \mathcal R^*_i$.
> > >
> > > As a result,
> > > $\left( \mathbf x^{(k)}\right)$ and $\mathbf x'$ belong to the set $\mathcal R^*$ on which $F_{\mathbf m^*}$ is continuous. It follows that $F(\mathbf x^{(k)}) = F_{\mathbf m^*}(\mathbf x^{(k)}) \downarrow F_{\mathbf m^*}(\mathbf x') = F(\mathbf x')$. Combining the fact that $F(\mathbf x^{(k)}) \downarrow F_{\mathbf m^*}(\bar{\mathbf x})$ by Step 1, we have $F(\mathbf x') = F_{\mathbf m^*}(\bar{\mathbf x})$.

---

> > > > ### Author Response · Authors · 2022-12-08
> > > > **Roadmap of Proof of Main Theorem (Theorem 4.4) with Indicated Modifications, and the Locally Optimal Convergence Rate with the Coercivity Assumption Part 2**
> > > >
> > > > Because $F_{\mathbf m^*} = F$ on the set $\left(\mathbf x \in \mathbb R^d \mid P(\mathbf x) = \mathbf m^* \right) \subseteq \mathcal R^*$, the optimality of $\bar{\mathbf x}$ indicates that
> > > >
> > > > $F(\mathbf x') = F_{\mathbf m^*}(\bar{\mathbf x}) \le \inf\limits_{\left(\mathbf x \in \mathbb R^d \mid P(\mathbf x) = \mathbf m^* \right)} F(\mathbf x) $.
> > > >
> > > > On the other hand, $\mathbf x' \in \left(\mathbf x \in \mathbb R^d \mid P(\mathbf x) = \mathbf m^* \right)$, so $F(\mathbf x')  \ge \inf\limits_{\left(\mathbf x \in \mathbb R^d \mid P(\mathbf x) = \mathbf m^* \right)} F(\mathbf x)$. Therefore, $F(\mathbf x')  = \inf\limits_{\left(\mathbf x \in \mathbb R^d \mid P(\mathbf x) = \mathbf m^* \right)} F(\mathbf x)$, proving that  $\mathbf x'$ achieves a local minimum of $F$.
> > > >
> > > >
> > > > **Step 3. (modification in the revised paper) proving any limit point $\mathbf x' \in \Omega$ is a critical point of $F$ under a mild condition**. Now we suppose the following mild condition holds, that is, $f_{\mathbf m_i^*}$ does not take the third case in (5) or (6) in the revised paper for all $i \in [d]$. Under such condition, we have $\mathcal R_i^* = \mathbb R$ for all $i \in [d]$. According to Step 1, $F(\mathbf x') = F_{\mathbf m^*}(\bar{\mathbf x}) = \min\limits_{\mathbf x \in \mathbb R^d} F_{\mathbf m^*}(\mathbf x)$. That is, $\mathbf x'$ achieves a global minimum for the convex optimization problem $\min\limits_{\mathbf x \in \mathbb R^d} F_{\mathbf m^*}(\mathbf x)$. The optimality of $\mathbf x'$ leads to $\mathbf 0 \in \tilde \partial F_{\mathbf m^*}(\mathbf x')$. Because $F_{\mathbf m^*} = F$ on the set $\left(\mathbf x \in \mathbb R^d \mid P(\mathbf x) = \mathbf m^* \right)$, it can be verified that $\mathbf 0 \in \partial F(\mathbf x')$ using the definition of the critical point.
> > > >
> > > > **Local Optimal Convergence Rate with the Coercivity Assumption**
> > > >
> > > > We respectfully point out that the local convergence rate $\mathcal O(\frac{1}{k^2})$ of PPGD is still the locally optimal rate under the mild coercivity assumption. It is noted that the objective function is widely assumed to be coercive in the nonconvex and nonsmooth optimization literature such as [A,B]. It can be observed from the Roadmap of Proof of Theorem 4.4 that the local convergence rate $\mathcal O(\frac{1}{k^2})$, which happens when $k > k_0$,  is in fact the convergence rate of applying a first-order method (PPGD) on the convex optimization problem $\min\limits_{\mathbf x \in \mathbb R^d, \mathbf x_i \in \mathcal R_i^*} F_{\mathbf m^*}(\mathbf x)$. As $\mathcal O(\frac{1}{k^2})$ is known to be the optimal rate of first-order method on smooth and convex objective function with Lipschitz continuous gradient, the local convergence rate of PPGD is optimal. In order to justify the optimality of such local convergence rate, one can repeat the same argument in [Nesterov (2004)] to find a coercive smooth and convex function, such as
> > > >
> > > > $g(\mathbf x) = \frac{L_g}{4} \left( \frac 12 \mathbf x_1^2 + \frac{1}{2} \sum_{i=1}^{2k}(\mathbf x_i - \mathbf x_{i+1})^2 + \frac 12 \mathbf x_{2k+1}^2 - \mathbf x_1 \right)$, with $\mathbf x \in \mathbb R^{2k+1}$ and $\nabla g$ being $L_g$-smooth,
> > > >
> > > > such that the convergence rate of any first-order optimization method on $\min_{\mathbf x \in \mathbb R^{2k+1}}g(\mathbf x)$ is lower bounded by $\mathcal O(\frac{1}{k^2})$.
> > > >
> > > >
> > > > **References**
> > > >
> > > > [A] Huan Li and Zhouchen Lin. Accelerated proximal gradient methods for nonconvex programming. NeurIPS 2015.
> > > >
> > > > [B] Qunwei Li, Yi Zhou, Yingbin Liang, and Pramod K. Varshney. Convergence analysis of proximal gradient with momentum for nonconvex optimization. ICML 2017.
> > > >
> > > > [Nesterov (2004)] Nesterov, Y., Introductory Lectures on Convex Optimization: A Basic Course, Kluwer Academic Publishers, Norwell, 2004.

---

> > > > > ### Comment · Reviewer_tNXt · 2022-12-13
> > > > > **Please be compact and to-the-point**
> > > > >
> > > > > Dear authors,
> > > > >
> > > > > Thanks for the detailed explanation of the proof idea. I only need a compact and to-the-point list of what has been modified and why. For example, though the bounded gradient assumption is removed, the newly introduced coercivity assumption implies a bounded level set, which allows you to still claim a bounded gradient and define the parameter $G$. This suffices for addressing the modification in Assumption 1. (I am not completely sure if the new coercivity assumption results in other modifications.)
> > > > >
> > > > > Nevertheless, although the explanation of the proof looks detailed, there are still many things unclear. For example, Theorem 4.3 in the revised version seems to correspond to Theorem 3.4 in the original version. But in the revised version, there is an additional upper bound for $s$ to satisfy, the last term in (13). Also, it is unclear to me why the original Lemma 3.1--Lemma 3.3 become Lemma 4.1 and Lemma 4.2 in the revised version.

---

> > > > > > ### Author Response · Authors · 2022-12-13
> > > > > > **All the Modifications in the Revised Paper (with a compact and to-the-point list)**
> > > > > >
> > > > > > Dear Reviewer tNXt,
> > > > > >
> > > > > > Thank you for your very helpful comment and the detailed example! Below is a complete yet compact and to-the-point list of modifications in the revised paper which lead to the results of the main theorem, including those unclear to you in your comments. When not specified explicitly, all the line numbers and lemma/theorem/equation numbers refer to those in the revised paper. Please kindly let us know if there are still unclear modifications or their effects and we will clarify them immediately.
> > > > > >
> > > > > > **Reorganization of Lemmas and Theorems.** Lemma 3.1-3.3 (in the original paper) are moved to the supplementary as Lemma B.4, B.2, and B.3 respectively, and Lemma 3.2-3.3 (in the original paper) are combined in Lemma 4.1 of the main paper. The motivation of such reorganization is to make presentation of the theoretical results clearer, and only the most important results are in the main paper. That is, the important result of the sufficient decline of the objective function (eq. (14)) when convex pieces of consecutive iterates change is stated in Lemma 4.1 of the main paper, which combines the sufficient decline results for two cases: the objective function is continuous (the original Lemma 3.2) or discontinuous (the original Lemma 3.3) at an endpoint. The original Lemma 3.1 is only needed in the proof of Lemma B.3, so it is stated as an auxiliary lemma (Lemma B.4) before the proof of Lemma B.3.
> > > > > >
> > > > > > It is noted that the sufficient decline in Lemma 4.1 is conditioned on the boundedness of the gradient, and such boundedness of gradient is guaranteed by Lemma 4.2. Theorem 4.3, which replaces Theorem 3.4 of the original paper, straightforwardly combines Lemma 4.1-4.2 to guarantee sufficient decline.
> > > > > >
> > > > > > **The Coercivity Assumption and its Associated Modifications.** Thank you for pointing out that the coercivity assumption leads to bounded gradient. To take advantage of the coercivity assumption, we have defined a new projection operator $P_{\mathbf x, R_0}$ in eq. (11) which replaces the original projection operator. This new projection operator ensures that $\mathbf w^{(k)}$ is always in the ($R_0$-sized) neighborhood of $\mathbf x^{(k)}$. It follows by this fact and the monotone nonincreasing of $\left(F(\mathbf x^{(k)})\right)$ that all the iterates $\left(\mathbf x^{(k)}\right)$ lie on a bounded level set, and $\left(\mathbf w^{(k)}\right)$ also lie on a bounded set $\mathcal L_{R_0}$ defined in eq. (12). The boundedness of $\left(\mathbf w^{(k)}\right)$ leads to the boundedness of the gradients $\nabla g(\mathbf w^{(k)})$, which in turn leads to Lemma 4.2 in the revised paper. It is noted that the new projection operator also satisfies eq. (56) (now eq. (73)) in the original paper when proving the convergence of PPGD in the main theorem (Theorem 4.4), which uses the contraction property of projection onto a closed convex set.
> > > > > >
> > > > > > **Modifications in the Negative-Curvature-Exploitation (NCE) Algorithm (Algorithm 2) and its Associated Changes.** Algorithm 2 has been modified by defining a new variable $\mathbf z'$ (line $5$) and adding the else-branch (line $16-17$); changing line $12-13$ in Algorithm 3 of the original paper to lines numbered the same. It is noted that lines $9-10$ in Algorithm 3 of the original paper are changed to a much simpler line $10$ because they are equivalent.
> > > > > >
> > > > > > Such modifications ensure that the NCE Algorithm can handle a convex piece comprising a single point, such as the convex piece $\left(0\right)$ which contains a single point $0$ for the $\ell^0$-norm penalty. In this case, $\mathbf z'$ stores such single point which is assigned to the next iterate $\mathbf x^{(k+1)}$. Because $\mathbf x^{(k+1)}$ could be set to $\mathbf z'$ instead of the previous iterate $\mathbf x^{(k)}$ or $\mathbf z^{(k+1)}$, it is not straightforward to see why the sufficient decline in Lemma 4.1 holds following the original proof. **To this end, we have added a new term (the last term in eq. (13)) as an additional upper bound for the step size $s$**. Lemma 4.1 states that, with this additional upper bound for $s$, the sufficient decline still holds. The proof of Lemma 4.1 is a revised version of the proof of Theorem 3.4 in the original paper. The main idea in the proof of Lemma 4.1 is based on the observation that $\mathbf z'$ is close to $\mathbf z^{(k+1)}$. The proof of the original Theorem 3.4 shows that sufficient decline happens when $\mathbf x^{(k+1)}$ is set to $\mathbf z^{(k+1)}$. Because $\mathbf z'$ is close to $\mathbf z^{(k+1)}$, it is proved that sufficient decline still holds with the additional upper bound for $s$.

---

> > > > > > > ### Comment · Reviewer_tNXt · 2022-12-14
> > > > > > > **Is equation 13 modified?**
> > > > > > >
> > > > > > > In the version of the paper I downloaded yesterday, there were five terms in the min mapping, and in the current version, there are only four terms. However, the author response still discusses about the "last term in eq. (13) as an additional upper bound." I wonder why there is such inconsistency.

---

> ### Author Response · Authors · 2022-12-06
> **Our Response to Reviewer tNXt (Reminder of Feedback)**
>
> Dear Reviewer tNXt,
>
> Thank you again for your comments in the original review. This is a gentle reminder of your feedback. As we are approaching the end of Discussion Stage 2, it would be very helpful for you to provide feedback to our response. The concerns in your original review are very specific which have been addressed by our response. In particular,
>
> "**(1) Completed Convergence Results of PPGD**" provides completed convergence results of PPGD with the optimality of the limit points of PPGD. Every limit point is a local minimum of the objective function, and also a critical point of the objective function under mild conditions.
>
> "**(2) The Claimed Optimality of $\mathcal O(\frac{1}{k^2})$**" clarifies the doubt regarding the local optimality of the convergence rate $\mathcal O(\frac{1}{k^2})$. We do not require the assumption that the convex smooth function $g$ is Lipschitz in the revised paper.
>
> "**(3) Efficient Computation of the Operator $P(\cdot)$**" explains that we do not need to assume $P(\cdot)$ is efficiently implementable. A simple and efficient way of computing $P$ is provided.
>
> "**Improved Presentation**" explains the significantly improved presentation in the revised paper. We have polished the paper according to your suggestions.

---

### Official Review · Reviewer_ofLD · 2022-10-23

**Confidence:** 4
**Correctness:** 3
**Technical Novelty And Significance:** 3
**Empirical Novelty And Significance:** Not applicable
**Recommendation:** 6

**Clarity, Quality, Novelty And Reproducibility:**

Technical quality: Good. I think this paper proposes a very interesting algorithm and is technically non-trivial. However, it seems to me that the paper is more like a preliminary draft that stacks the definitions and results with dense notations.

Clarity, Quality: Poor. The overall presentation, grammar and writing are below average and highly nonsmooth. I suggest the authors further polish the details of the paper substantially, add more background and motivation to the introduction.

**Strength And Weaknesses:**

Strength:

- This paper considers composite optimization problems that involves piecewise convex regularizers, this covers a broad class of machine learning applications.

- The algorithm design is substantially different from the existing ones. In particular, both the new projection operator and the Negative-Curvature-Exploitation subroutine exploit the special piecewise structure of the regularizer. This is why the algorithm can achieve a good numerical performance.

-The analysis is not based on the general KL geometry that can be very loose at the beginning of the optimization process. Instead, the authors analyze how the iterates enter different convex pieces and eventually stay within a certain piece and achieve the optimal rate asymptotically.

Weakness:

- The assumption 1 assumes bounded gradient, which is not required by standard accelerated methods. Also, assuming a bounded gradient over the entire space may be unrealistic.

- It is not clear to me why the authors want to construct the surrogate function $f_m$ in that specific way. Moreover, it is not clear what is the intuition and motivation to construct it in the way as illustrated in Figure 2.

- In eq(10), how is the index $i$ specified?

- Assumption 2 is justified for the $\ell_1$ penalty function when $\lambda>G$. But in reality, the gradient norm upper bound $G$ can be very large. In Lemma 3.1-3.3, what is $s$? I did not see a definition for it.

**Summary Of The Paper:**

This paper studies the composite optimization problem $g(x) + h(x)$ where $g$ is convex smooth, and $h$ can be nonconvex and nonsmooth but is separable and piecewise convex. By leveraging the special structure of $h$, the authors proposed an interesting proximal type algorithm that asymptotically converges at the optimal rate. The algorithm introduces a special projection operator that finds the nearest endpoint, and also involves a Negative-Curvature-Exploitation subroutine. The main idea is to show that the iterates eventually stay within a fixed convex piece.

**Summary Of The Review:**

see above

---

> ### Author Response · Authors · 2022-11-20
> **Response to Reviewer ofLD Part 1**
>
> We appreciate the review and the suggestions in this review. The raised issues are addressed below.
>
> (1) **Assumption about Bounded Gradient**
>
> The assumption about bounded gradient, $\sup_{\mathbf x \in \mathbb R^d} \vert\vert g(\mathbf x) \vert\vert \le G$, is removed. Instead, we adopt the widely used assumption that the objective function $F$ is coercive, that is, $F(\mathbf x) \to \infty$ when $\vert\vert \mathbf x \vert\vert \to \infty$, and $\inf_{x \in \mathbb R^d} F(x) > -\infty$. The objective $F$ is widely assumed to be coercive in the nonconvex and nonsmooth optimization literature such as [A,B]. Such coercive assumption replaces the original assumption that $\sup_{\mathbf x \in \mathbb R^d} \vert\vert g(\mathbf x) \vert\vert \le G$ in Assumption 1(a) of the revised paper. Under such coercive assumption and a novel projection operator defined in eq. (11), we derive the upper bound $G$ (defined in eq. (12)) such that $\sup_{k \ge 1} \vert\vert \nabla g(\mathbf w^{(k)}) \vert\vert \le G$ in a new lemma, Lemma 4.2. This new bound $G$ in Lemma 4.2 is much smaller than the original bound for $\sup_{\mathbf x \in \mathbb R^d} \vert\vert g(\mathbf x) \vert\vert$ which is the supremum of $\vert\vert g(\mathbf x) \vert\vert$ over the entire $\mathbb R^d$.
>
> (2) **The Intuition and Motivation to Construct Surrogate Functions, and Improved Presentation**
>
> Detailed description about the intuition and motivation to construct surrogate functions has been added to the revised paper. We have also considerably improved the presentation with more background and motivation.
>
> In particular, there is a new Section 2 in the revised paper, “Roadmap to Fast Convergence by PPGD”, which presents the intuition and motivation of surrogate functions and an intuitive roadmap to the fast convergence of PPGD. Section 2 is copied below for your convenience.
>
> Two essential components of PPGD contribute to its fast convergence rate. The first component, a combination of a carefully designed Negative-Curvature-Exploitation algorithm and a new projection operator, decreases the objective function by a positive amount when the iterates generated by PPGD transit from one convex piece to another. As a result, there can be only a finite number of such transitions. After finite iterations, all iterates must stay on the same convex pieces. Restricted on these convex pieces, problem (1) is convex.  The second component, which comprises $M$ surrogate functions, naturally enables that after iterates reach their final convex pieces, they are operated in the same way as a regular APG does so that the convergence rate of PPGD locally matches the Nesterov's optimal rate achieved by APG. Restricted on each convex piece, the piecewise convex function $f$ is convex. Every surrogate function is designed to be an extension of this restricted function to the entire $\mathbb R$, and PPGD performs proximal mapping only on the surrogate functions.
>
> We have also introduced the intuition and motivation of surrogate functions in Section 3.1 of the revised paper at a technical level, which is presented below.
>
> The key idea of each surrogate function $f_m$ is to extend the domain of $f_m$ from $\mathcal R_m$ to $\mathbb R$ with the simplest structure, that is, $f_m$ is linear outside of $\mathcal R_m$. More concretely, if the right endpoint $q = q_{m}$ is not $+\infty$ and $f$ is continuous at $q$, then $f_m$ extends $f \vert_{\mathcal R_m}$ such that $f_m$ on $(q,+\infty)$ is linear. Similar extension applies to the case when $q = q_{m-1}$ is the left endpoint of $\mathcal R_m$.
>
> We have scrutinized the paper and fixed all the grammatical issues and typos in the revised paper.
>
> (3) **Remaining Issues**
>
> We have removed the typo in the PPGD algorithm so that the new eq. (10) does not have index $i$.
>
> Regarding the issue that the original gradient bound $G$ in $\sup_{\mathbf x \in \mathbb R^d} \vert\vert g(\mathbf x) \vert\vert \le G$ can be very large, it is noted in part (1) of this response that we do not need $G$ to bound $\sup_{\mathbf x \in \mathbb R^d} \vert\vert g(\mathbf x) \vert\vert$ in the revised paper. Instead, based on the widely adopted coercive assumption in Assumption 1(a), we derive a much smaller bound $G$ (defined in eq. (12)) such that $\sup_{k \ge 1} \vert\vert \nabla g(\mathbf w^{(k)}) \vert\vert \le G$. Therefore, in the example of capped-$\ell^1$ penalty which justifies Assumption 2, we only need $\lambda$ to be larger than the much smaller $G$ defined in eq. (12). We have broad applications, especially in sparse learning, where such large $\lambda$ is used. For example, when one needs to use capped-$\ell^1$ to approximate $\ell^0$-norm and achieve low statistical estimation error, then $\lambda$ is allowed to be larger than $G$ in eq. (12). In [Zhang, 2010, Zhang 2013], such large $\lambda$ is used with provable low statistical estimation error.
>
> $s$ is specified as the step size in the revised paper.

---

> > ### Author Response · Authors · 2022-11-20
> > **Response to Reviewer ofLD Part 2 (References)**
> >
> > **References**
> >
> > [A] Huan Li and Zhouchen Lin. Accelerated proximal gradient methods for nonconvex programming. NeurIPS 2015.
> >
> > [B] Qunwei Li, Yi Zhou, Yingbin Liang, and Pramod K. Varshney. Convergence analysis of proximal gradient with momentum for nonconvex optimization. ICML 2017.
> >
> > [Zhang, 2010] Tong Zhang. Analysis of multi-stage convex relaxation for sparse regularization. The Journal of Machine Learning Research, 11:1081–1107, 2010.
> >
> > [Zhang, 2013] Tong Zhang. Multi-stage convex relaxation for feature selection. Bernoulli, 19(5B):2277–2293, 11 2013.

---

### Official Review · Reviewer_V486 · 2022-10-26

**Confidence:** 4
**Correctness:** 3
**Technical Novelty And Significance:** 3
**Empirical Novelty And Significance:** 2
**Recommendation:** 6

**Clarity, Quality, Novelty And Reproducibility:**

This paper is well written. The technique seems novel and nontrivial. But I did not check all the proofs.

**Strength And Weaknesses:**

Nonconvex nonsmooth optimization is important but challenging setting. This paper contributes to this area by identifying tractable problem classes and introducing an algorithm with rate estimation. The construction is nontrivial and seems novel. They also demonstrate the empirical performance on real-world data.

My main concerns are as follows:
* The O(1/k^2) rate in Thm 3.5 should not be compared with that of Nesterov's optimal rate for the smooth convex function. The point is that the rate in Thm 3.5 is only a local rate, while Nesterov's is a global one. In other words, we generally cannot quantify how large the k_0 is, which could be arbitrarily large but finite. But I'm not strongly against this point as this type of local analysis is common in both convex/nonconvex nonsmooth problem, see:

[r1] Manifold identification in dual averaging for regularized stochastic online learning. JMLR 12.
[r2] Are we there yet? Manifold identification of gradient-related proximal methods. AISTATS 19.
[r3] Computing D-Stationary Points of rho-Margin Loss SVM. AISTATS 20.

* On "without KL": This paper claims their convergence results are better in the sense that they don't use the KL inequality. But it seems (if I understand correctly), existing analysis, e.g., (Li et al. 2017), already show convergence (of stationarity measure) without KL (use Eq. (21+22) in their paper). The key point here is that we use the regularity from KL to show the convergence of {x_t}. But this paper only shows the convergence of F(x_k) - F(x_*) with that of {x_t}, which may not be compared with the existing sequential convergence results.

Minor:
* Section 2.1: "both g and h are" ???
* Above Eq.(6): "by lsc of f". I did not find the assumption that f is lsc.
* Eq.(9): "\nabla g(w^{(k)}"
* Lemma B.1 is Rockafellar-Wets (Theorem 10.1).
* Do you really need to introduce the notions of Frechet and limiting subdifferential? It seems these notion is only used above Eq.(19). But here the prox operator is for convex problem. Maybe convex subdifferential suffices.

**Summary Of The Paper:**

This paper considers the optimization of a class of nonconvex nonsmooth problems which is the sum of a smooth convex function and a separable piecewise convex regularization term. The challenging part is mainly from the regularization term, which could be nonconvex, nonsmooth, and even discontinuous. They introduce a variant of PGD called PPGD and show that PPGD has a local sublinear convergence rate.

The main observation is that, for a one-dimensional piecewise convex function, there exist finite intervals on which the whole function is convex and smooth. The authors carefully introduce a series of regularity conditions (in Assumption 1 and 2) to ensure all critical points (in the limiting sense) cannot be at the breaking point between different pieces. Then, by a manifold identification-type argument, there cannot be infinite piece changing in the iteration, which restricts the analysis to a local convex smooth problem. The local rate follows from existing work.

**Summary Of The Review:**

This paper considers the optimization of a class of nonconvex nonsmooth problems which is the sum of a smooth convex function and a separable piecewise convex regularization term. My main concerns are (1) The O(1/k^2) rate in Thm 3.5 should not be compared with that of Nesterov's optimal rate for the smooth convex function; (2) The convergence without KL results may not be compared with the existing sequential convergence results, where the latter is much stronger.

---

> ### Author Response · Authors · 2022-11-20
> **Response to Reviewer V486**
>
> We appreciate the review and the suggestions in this review. The raised issues are addressed below.
>
> (1) **The $\mathcal O(\frac{1}{k^2})$ Convergence Rate**
>
> We have revised our statement regarding the locally optimal convergence rate of PPGD based on your suggestion, and we appreciate your suggestion. In particular, it is emphasized in the revised paper that the convergence rate of $\mathcal O(\frac{1}{k^2})$ is a locally optimal rate in the sense that it holds when $k \ge k_0$ for a  finite $k_0$, for example, in Section 1.2 of the revised paper. We respectfully point out that the local rate $\mathcal O(\frac{1}{k^2})$ is locally optimal, because all the iterates $\left( \mathbf x^{(k)}\right)$ for $k > k_0$ are on the same convex pieces indexed by $\mathbf m^*$, and restricted on such convex pieces the objective $F$ is smooth and convex where $\mathcal O(\frac{1}{k^2})$ is known to be an optimal rate.
>
> (2) **Convergence with and without the KL Property**
> We respectfully point out that although Theorem 1 in (Li et al. 2017) proves that $F(\mathbf x^{(k)}) \overset{k \to \infty}{\to}F(\mathbf x^*)$ where $\mathbf x^*$  is a critical point of the sequence $\left( \mathbf x^{(k)}\right)$ without the KL property, there is no actual convergence rate for such convergence.  All the convergence rates of function values in (Li et al. 2017) are proved under the KL property, and the KL property is not used to prove the convergence of the sequence $\left( \mathbf x^{(k)}\right)$ (and there is no convergence rate of this sequence) in (Li et al. 2017). In fact, using only eq. (21) and eq. (22) in (Li et al. 2017) cannot derive any convergence rate. In the proof of Theorem 2 of (Li et al. 2017), eq. (21) and eq. (22) are used to establish one condition required by the Uniformized KL property (defined in their Definition 5), that is, $\mathbf x^{(k)} \in \left( \textup{dist}_{\Omega}(\mathbf x^{(k)}) < \epsilon, F^* < F(\mathbf x^{(k)}) < F^* + \delta \right)$. This condition combined with the KL property leads to eq. (23) in in (Li et al. 2017), which then indicates the convergence rate of the function values.
>
> As a result, to the best of our knowledge, this paper is still the first work which presents a locally fast (sublinear) convergence rate for an important class of nonconvex nonsmooth problems without the KL property.
>
> **Remaining Issues**
>
> Rockafellar-Wets (Theorem 10.1) is cited in Lemma B.1 of the revised paper. We have fixed all the grammatical issues and typos in the revised paper.
>
> **Regarding the usage of Frechet limiting-subdifferential**  A critical point is defined as a point whose Frechet limiting-subdifferential includes $\mathbf 0$. Our main theorem, Theorem 4.4 of the revised paper, shows that a limit point can be a critical point of the objective function eq. (1) under mild conditions. It is noted that such limit point, which is also a critical point of the objective function, can be some endpoint where $f$ is continuous. For example, one coordinate of such limit point can be $b$ which is an endpoint of the capped-$\ell^1$ penalty in Figure 1. In this case, such limit point does not have the regular convex subdifferential because $f$ is not convex at this point (for example, $b$). However, it is proved in Theorem 4.4 that such limit point has Frechet limiting-subdifferential which includes $\mathbf 0$.

---

### Official Review · Reviewer_Dzxh · 2022-11-01

**Confidence:** 3
**Correctness:** 3
**Technical Novelty And Significance:** 2
**Empirical Novelty And Significance:** 2
**Recommendation:** 3

**Clarity, Quality, Novelty And Reproducibility:**

Algorithm 3 is incomplete (and is also not clearly/concisely written), so it is difficult to verify whether the related analysis in the paper is correct.

**Strength And Weaknesses:**

**Strength**
- The proposed method has the accelerated rate for some nonsmooth and nonconvex problems without the KL property, which is not known before.

**Weakness**
- Restrictive assumption: Although the analysis does not require the standard KL property for the nonsmooth and nonconvex analysis, the assumptions in this paper is not so general as expected from the title. The nonsmooth regularization should be separable, piecewise convex, and proximal friendly.
- Incomplete Algorithm 3: Output of Algorithm 3 when flag is true is not stated, so it is not clear how $x^{(k+1)}$ is chosen. This makes it difficult to verify following analysis in Lemma 3.1 - 3.3. Also, a brief explanation of each step of this incomplete and rather complicated method is given at the end of Section 2, but it does not seem to be sufficient to readers.
- $P(x) \neq P(y)$ indicates $x,y$ are on different convex pieces (The authors claim that this is important in the analysis): If $x$ and $y$ are on the same convex piece and $x$ is an endpoint at which $f$ is continuous, then I think it is possible to have $P(x) \neq P(y)$.
- Theorem 3.5: $x^*$ is a limit point of Algorithm 2, but it is not stated whether Algorithm 2 converges to any desirable minimum point. This makes the accelerated rate analysis yet not so much interesting.

**Miscellaneous**
- After (1): How about letting the readers know that $g$ is convex here, although this is later stated.
- Define the notation $[M]$
- Assumption 1(b): $f$ is "differentiable"; second assumption is missing.
- (4): This came from Theorem 3.5, and the norm is not necessary; $x^*$ is not defined
- "a" new perspective
- Assuming that both $g$ and $h$ are ???
- $q = q_{m-1}$ is the "left" endpoint
- According to "Definition" 1,
- $\nabla g (w^{(k)}$"$)$"
- Algorithm 3: awkward line breaking at line 10

**Summary Of The Paper:**

This paper constructs an efficient first-order method, named PPGD, that solves a nonconvex and nonsmooth problem, especially with a nonsmooth (separable) _piecewise convex_ regularization term, such as an indicator penalty, a capped-$\ell_1$ penalty and a leaky capped-$\ell_1$ penalty. One notable contribution is that the analysis does not involve the KL property, unlike other existing analysis on nonconvex and nonsmooth optimization. In addition, the PPGD has an $O(1/k^2)$ rate that is the optimal rate of first-order methods for smooth and convex optimization.

**Summary Of The Review:**

This paper came up with a new method for minimizing a structured nonconvex and nonsmooth problem with a nonsmooth (separable) piecewise convex regularization term, built upon the monotone accelerated proximal gradient descent method (Algorithm 1). Handling such regularization term seems novel and interesting, but the main Algorithm 3 is missing some lines, which makes it difficult to verify its related lemmas. In addition, there are some issues in the authors' claims state above.

---

> ### Author Response · Authors · 2022-11-19
> **Response to Reviewer Dzxh Part 1**
>
> We appreciate the review and the suggestions in this review. The raised issues are addressed below.
>
> (1) **Relaxed Assumptions**
>
> We have provided considerably novel analysis and extension in the revised paper which relax three major conditions in the main assumption, Assumption 1. These relaxations are explained below.
>
> (a) The assumption about bounded gradient, $\sup_{\mathbf x \in \mathbb R^d} \vert\vert \nabla g(\mathbf x) \vert\vert \le G$, is removed. Instead, we adopt the widely used assumption that the objective function $F$ is coercive, that is, $F(\mathbf x) \to \infty$ when $\vert\vert \mathbf x\vert\vert \to \infty$, and $\inf_{x \in \mathbb R^d} F(x) > -\infty$. The objective $F$ is widely assumed to be coercive in the nonconvex and nonsmooth optimization literature such as [A,B]. Under such coercive assumption and a novel projection operator defined in eq. (11), we derive the upper bound $G$ (defined in eq. (12)) such that $\sup_{k \ge 1} \vert\vert \nabla g(\mathbf w^{(k)}) \vert\vert \le G$. This new bound $G$ is derived based on the popular and widely satisfied coercive assumption, and it is much smaller than the original bound for $\sup_{\mathbf x \in \mathbb R^d} \vert\vert \nabla  g(\mathbf x) \vert\vert$ which is the supremum of $\vert\vert \nabla  g(\mathbf x) \vert\vert$ over the entire $\mathbb R^d$.
>
> (b) The assumption that $f$ is piecewise convex can be relaxed to a weaker one detailed in Remark 4.6 of the revised paper. More concretely, we only need $f$ to be convex on the final convex pieces $\mathcal R_{\mathbf m_i^*}$ for all $i \in [d]$. The reason is that, by Theorem 4.4, there exists a finite $k_0$ such that the iterates $(\mathbf x^{(k)})$ for $k > k_0$ are on the convex pieces indexed by $\mathbf m^* \in \mathbb N^d$. All the statements of Theorem 4.4, the major theorem of this paper, are still true if $f$ is convex on the convex pieces $\mathcal R_{\mathbf m_i^*}$ for all $i \in [d]$ and $f$ is not convex on other convex pieces.
>
> (c) The proximal mapping in Assumption 1(c) does not need to have a closed-form solution, that is, $f$ does not need to be friendly for proximal mapping. Inexact PPGD is presented in Section C of the supplementary, where the same order of convergence rate as PPGD is proved with inexact proximal mapping for inexact PPGD. The optimality of limits points of PPGD still holds for the limit points of inexact PPGD.
>
>  **Significance of PPGD** As a result of the above major relaxations, the proposed PPGD enjoys the same order of convergence rate and the same optimality of the limit points the iterates of PPGD converge to under much less restrictive assumptions which are mostly standard in nonconvex and nonsmooth optimization literature for separable regularizers. We would like to emphasize that this paper, to the best of our knowledge, presents the first work with provable fast convergence rate (sublinear rate) on an important class of nonconvex and nonsmooth problems without the assumption about the $KL$ property, while locally matching the Nesterov's optimal convergence rate. Our assumption admits popular nonconvex and nonsmooth regularizers such as the indicator penalty, capped-$\ell^1$ penalty, leaky capped-$\ell^1$ penaly, and the $\ell^0$-norm penalty. More importantly, we propose the novel Negative-Curvature-Exploitation algorithm. To the best of our knowledge, this is the first algorithm that guarantees that proximal gradient method can make progress in decreasing the objective value by exploring the natural “negative curvature” (formally defined in Assumption 1(d)) in the aforementioned popular nonconvex and nonsmooth regularizers. It is also worthwhile to point out that previous works usually use algorithms analyzed under the KL assumption to optimize objective functions with the aforementioned popular regularizers. For example, [A] conducts numeral experiments on sparse logistic regression with capped-$\ell^1$ penalty, which is indeed covered by our PPGD analysis without the KL assumption. This paper shows that such sparse logistic regression can enjoy sublinear convergence rate without the KL assumption.
>
> The title of the revised paper has been changed to "Projective Proximal Gradient Descent for A Class of Nonconvex Nonsmooth Optimization Problems: Fast Convergence Without Kurdyka-Lojasiewicz (KL) Property". It is also noted that there is a simple yet novel change to the projection operator in eq. (8) of the PPGD Algorithm 1 in the revised paper. We found that this minor change does not affect our numerical results.

---

> > ### Author Response · Authors · 2022-11-19
> > **Response to Reviewer Dzxh Part 2**
> >
> > (2) **Completed Algorithm**
> >
> > We have completed and simplified the Negative-Curvature-Exploitation algorithm, which is now Algorithm 2 of the revised paper. We also give an intuitive description of this algorithm in the second paragraph under Algorithm 2, and it is copied here for your convenience.
> > The idea of of Algorithm 2 is to use $\mathbf z^{(k+1)}$ as a probe for the next convex pieces that the current iterate $\mathbf x^{(k)}$ should transit to. Compared to the regular APG described in Algorithm 4 in Section 3.2 of the supplementary, the projection of $\mathbf u^{(k)}$ onto a closed convex set is used to compute $\mathbf z^{(k+1)}$. This is to make sure that $\mathbf u^{(k)}$ is ``dragged'' back to the convex pieces of $\mathbf x^{(k)}$, and the only variable that can explore new convex pieces is $\mathbf z^{(k+1)}$. We have a novel Negative-Curvature-Exploitation (NCE) algorithm described in Algorithm 2 which decides if PPGD should update the next iterate $\mathbf x^{(k+1)}$. In particular, if $\mathbf z^{(k+1)}$ is on the same convex pieces as $\mathbf x^{(k)}$, then $\mathbf x^{(k+1)}$ is updated to $\mathbf z^{(k+1)}$ only if $\mathbf z^{(k+1)}$ has smaller objective value. Otherwise, the NCE algorithm carefully checks if any one of the two sufficient conditions can be met so that the objective value can be decreased. If this is the case, then $\textup{Flag}$ is set to True and $\mathbf x^{(k+1)}$ is updated by $\mathbf z^{(k+1)}$, which indicates that $\mathbf x^{(k+1)}$ transits to convex pieces different from that of $\mathbf x^{(k)}$. Otherwise, $\mathbf x^{(k+1)}$ is set to the previous value $\mathbf x^{(k)}$. The two sufficient conditions are checked in line $8$ and $10$ of Algorithm 2.
> >
> > (3) **$P(x) \neq P(y)$ indicates that $x$ and $y$ are on different convex pieces**
> >
> > We have revised the endpoints of the convex pieces in the second last paragraph of page 1 of the revised paper, so that any point in $\mathbb R$ only belongs to one convex piece without affecting all the theoretical and empirical results. As a result, $P(x) \neq P(y)$ indicates that $x$ and $y$ are on different convex pieces.
> >
> > (4) **Completed Convergence Results of PPGD**
> >
> > The major theorem, Theorem 4.4 in the revised paper, presents the completed convergence results of PPGD with the optimality of the limit points the iterates of PPGD converge to. Theorem 4.4 is copied below for your convenience which is followed by its explanation.
> >
> > **Theorem 4.4**
> >
> > Suppose the step size $s < \min\set{s_1,\frac{\epsilon_0}{L_g (G+\sqrt{d}F_0)},\frac{1}{L_g}}$ with $s_1$ defined by eq. (16),  Assumption 1 and Assumption 2 hold. Then there exists a finite $k_0 \ge 1$ such that the following statements hold. (1) $P(\mathbf x^{(k)}) = \mathbf m^*$ for some $\mathbf m^* \in \mathbb N^d$ for all $k > k_0$. (2) Let $\Omega \coloneqq
> > \left( \mathbf x \mid \mathbf x \textup{ is a limit point of }  (\mathbf x^{(k)}), P(\mathbf x) = \mathbf m^* \right)$ be the set of all limit points of the sequence $\left(\mathbf x^{(k)}\right)$ generated by Algorithm 1lying on the convex pieces indexed by $\mathbf m^*$. Then for any $\mathbf x^* \in \Omega$, $F(\mathbf x^*) = \inf\limits_{\left(\mathbf x \in \mathbb R^d \mid P(\mathbf x) = \mathbf m^* \right)} F(\mathbf x)$, and
> >
> > $F(\mathbf x^{(k)})-F(\mathbf x^*) \le \frac{4}{k^2} U^{(k_0)} \quad (18)$
> >
> > for all $k > k_0$, where $U^{(k_0)} \coloneqq \left( \frac{1}{2s} \lvert\lvert (t_{k_0-1}-1) \mathbf x^{(k_0-1)} - t_{k_0-1} \mathbf z^{(k_0)} + \mathbf x^* \rvert\rvert_2^2 + t_{k_0-1}^2 ( F(\mathbf x^{(k_0)})-F(\mathbf x^*) ) \right)$. Moreover, if $f_{\mathbf m_i^*}$ does not take the third case in (5) or (6) for all $i \in [d]$, then $\mathbf x^*$ is a critical point of $F$, that is, $0 \in \partial F(\mathbf x^*)$.
> >
> > Theorem 4.4 will be explained by Part 3 of this response.

---

> > > ### Author Response · Authors · 2022-11-19
> > > **Response to Reviewer Dzxh Part 3**
> > >
> > > **Explanation of the Results of Theorem 4.4**
> > >
> > > Theorem 4.4 states that after a finite number ($k_0$) of iterations, all iterates $\left(\mathbf x^{(k)}\right)$ with $k > k_0$ are on the same convex pieces indexed by $\mathbf m^*$, and any limit point $\mathbf x^*$ of the sequence $ (\mathbf x^{(k)})$ on such convex pieces has the following nice properties. (1) $\mathbf x^*$ is a local minimum of the objective function $F$ on the convex pieces indexed by $\mathbf m^*$, that is, $F(\mathbf x^*) = \inf\limits_{\left(\mathbf x \in \mathbb R^d \mid P(\mathbf x) = \mathbf m^* \right)} F(\mathbf x)$. (2) $F(\mathbf x^{(k)})$ converges to $ F(\mathbf x^*) $ at a rate of $\mathcal O(1/k^2)$, which locally matches the Nesterov’s optimal convergence rate of first-order methods on smooth and convex objective function with Lipschitz continuous gradient. (3) $\mathbf x^*$ is a critical point of $F$ if $f_{\mathbf m_i^*}$ does not take the third case in (5) or (6) for all $i \in [d]$, that is, the surrogate function $f_{\mathbf m_i^*}$ does not have “jump-up” discontinuity, or formally, both $f(q_{\mathbf m_i^*}) < \lim_{y \to q_{\mathbf m_i^*}^+} f(y)$ and $f(q_{\mathbf m_i^*-1}) < \lim_{y \to q_{\mathbf m_i^*-1}^-} f(y)$ do not happenfor all $i \in [d]$. This scenario happens for all continuous $f$, for example, capped-$\ell^1$ or leaky capped-$\ell^1$ penalty functions.
> > >
> > > In addition to the above revisions, we have also fixed all the grammatical issues and typos in the revised paper.
> > >
> > > **References:**
> > > - [A] Huan Li and Zhouchen Lin. Accelerated proximal gradient methods for nonconvex programming. NeurIPS 2015.
> > > - [B] Qunwei Li, Yi Zhou, Yingbin Liang, and Pramod K. Varshney. Convergence analysis of proximal gradient with momentum for nonconvex optimization. ICML 2017.

---

> ### Author Response · Authors · 2022-12-06
> **Our Response to Reviewer Dzxh (Reminder of Feedback)**
>
> Dear Reviewer Dzxh,
>
> Thank you again for your comments in the original review. This is a gentle reminder of your feedback. As we are approaching the end of Discussion Stage 2, it would be very helpful for you to provide feedback to our response. The concerns in your original review are very specific which have been addressed by our response. In particular,
>
> "**(2) Completed Algorithm**" explains the completed Negative-Curvature-Exploitation algorithm with detailed explanation.
>
> "**(1) Relaxed Assumptions**" addresses the concern of restrictive assumptions with three relaxed assumptions in our Main Assumption 1.
>
> "**(3) $P(x) \neq P(y)$ indicates that $x$ and $y$ are on different convex pieces**" addresses the concern about $P$.
>
> "**(4) Completed Convergence Results of PPGD**" provides completed results about the optimality of the limit points of PPGD. More concretely, every limit point is a local minimum of the objective function, and also a critical point of the objective function under mild conditions.

---

### Author Response · Authors · 2022-12-09
**Request for reviewers' feedback to our response**

Dear Reviewers,

Thank you for your time reviewing this paper. We thank reviewer tNXt for the request for pointing out the modifications in the revised paper which prove the completed convergence results of PPGD (Theorem 4.4 in the revised paper). We responded to such request with intuition behind our proof and detailed roadmap of the proof.

As we are approaching the end of Discussion Stage 2, this is a gentle reminder of your feedback. We would like to let you know again that the concerns in all the reviews, especially the technical concerns from reviewer Dzxh and reviewer tNXt, have been addressed in our individual response to each reviewer and also summarized in "Summary of Our Responses and Revisions". Because the current ratings of this paper are largely based on the very specific technical concerns which have already been addressed substantially, your feedback is particularly important and greatly appreciated.

We also would like to let reviewer Dzxh and other reviewers know that we are more than happy to provide more detailed explanations (intuition and roadmap) for our algorithms, theoretical results, and the proofs whenever any reviewer needs them.

We look forward to your feedback to our response.

Thank you for your time!

Best Regards,

Authors

---

### Decision · Program_Chairs · 2023-01-20

**Decision:**

Accept: poster

**Justification For Why Not Higher Score:**

The work requires another revision that clearly puts their contribution within the context of existing works (from the title to assumptions). Moreover, a bit more clarification is needed on the limit points of the algorithm and what their rate implies.

**Justification For Why Not Lower Score:**

The authors obtain an accelerated rate for a class of nonsmooth and nonconvex problems without the KL property, which was not available before (certainly not without the weak-convexity assumption).

**Metareview: Summary, Strengths And Weaknesses:**

The authors obtain an accelerated rate for a class of nonsmooth and nonconvex problems without the KL property, which was not available before (certainly not without the weak-convexity assumption).

The results are actually not as general as the title implies due to the assumptions, such as the regularizer being piecewise convex, separable as well as proximal friendly. There are some clarifications issues that the reviewers pointed out. The authors managed to fix some of them to improve their presentation and the current version is significantly better.


**Note From Pc:**

if the above contains the word "oral" or "spotlight" please see: "oral" presentation means -> notable-top-5% and "spotlight" means -> notable-top-25%. As stated in our emails, we are disassociating presentation type from AC recommendations